# Do Data Valuations Make Good Data Prices?

## Abstract

As large language models increasingly rely on external data sources, compensating data contributors has become a central concern. But how should these payments be devised? We revisit data valuations from a *market-design perspective* where payments serve to compensate data owners for the *private* heterogeneous costs they incur for collecting and sharing data. We show that popular valuation methods—such as Leave-One-Out and Data Shapley—make for poor payments. They fail to ensure truthful reporting of the costs, leading to *inefficient market* outcomes. To address this, we adapt well-established payment rules from mechanism design, namely Myerson and Vickrey-Clarke-Groves (VCG), to the data market setting. We show that Myerson payment is the minimal truthful mechanism, optimal from the buyer's perspective. Additionally, we identify a condition under which both data buyers and sellers are utility-satisfied, and the market achieves efficiency. Our findings highlight the importance of incorporating incentive compatibility into data valuation design, paving the way for more robust and efficient data markets. Our data market framework is readily applicable to real-world scenarios. We illustrate this with simulations of contributor compensation in an LLM based retrieval-augmented generation (RAG) marketplace tasked with challenging medical question answering.

## 1 Introduction

The emergence of large language models (LLMs) has placed data at the heart of technological and societal advancement. As concerns mount that the availability of data may not keep pace with the rapid growth of model sizes (Villalobos et al., 2024), the ability to source high-quality data has become a critical factor for the success of LLM companies. Currently, web-scale data crawling is conducted with little regard for data provenance, often leading to copyright infringement that impacts content owners. This has resulted in a growing number of copyright lawsuits, such as those documented by New York Times v. OpenAI (2023); Concord Music Group v. Anthropic (2023).

Increasingly, content creators are choosing to opt out of contributing their work for AI training (Longpre et al., 2024; Fan et al., 2025). To encourage participation from data owners, there is a rising need to design an efficient data-trading market. Data owners should be fairly compensated for the use of their data, taking into account factors like the intrinsic cost of data generation, for example, it is more costly to gather expert-curated data than synthetic data (Figure 1). However, these costs are private - only the data collectors themselves know the effort and cost they put in. A well designed data marketplace could nevertheless discover these heterogeneous costs, match high-cost data collectors only with buyers who derive a high-value from the dataset, and extract appropriately high compensation. It is unclear if existing approaches to data pricing achieve such goals.

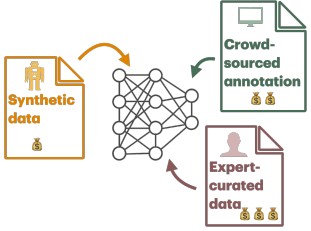

Figure 1: Data sources have different *private* unit costs

Existing data valuation methods for machine learning primarily focus on interpretability and guiding data collection, often neglecting market dynamics. In a market scenario, data owners (sellers) can misreport data-related costs to maximize their compensation, while data buyers strive to improve model performance at minimal cost. When valuation methods such as Leave-One-Out (Weisberg & Cook, 1982) and Data Shapley (Ghorbani & Zou, 2019) are directly used as pricing rules in data

markets, we point out that data sellers are incentivized to misreport their true costs, resulting in inefficient data market collaboration.

To address the challenge of untruthful reporting, we draw on well-established payment mechanisms from game theory, specifically, the Myerson payment rule (Myerson, 1981) and the Vickrey–Clarke–Groves (VCG) mechanism (Vickrey, 1961; Clarke, 1971; Groves, 1973), and adapt them to our data trading framework. Our analysis shows that: (1) the Myerson payment rule yields the minimum possible payment, making it optimal from the buyer's standpoint; and (2) when allocations are made to maximize overall market welfare in an unconstrained setting, the VCG and Myerson payments coincide. Additionally, we demonstrate that when buyers' utility functions are subadditive, total payments can be distributed across buyers while maintaining individual rationality for all participants. These findings highlight the crucial role of game-theoretic principles in designing data market payment schemes.

Our data market modeling framework can be readily extended to real-world applications, ranging from simple mean estimation to optimizing data mixtures for LLM pretraining, to compensating content contributors in retrieval-augmented generation (RAG) systems. For the RAG market, the payments can be easily calculated by modifying the existing post-retrieval reranking process, showing great potential for seamless integration into current pipelines while enabling fair and truthful compensation for document owners.

## 2 RELATED WORK

**Data Valuation Methods.** Data valuation has gained growing popularity in machine learning applications, mainly for the purposes of explainability and addressing algorithmic fairness (Pruthi et al., 2020; Liang et al., 2022), guiding high-quality data selection (Chhabra et al., 2024; Yu et al., 2024). Among them there are two primary categories methods – Shapley value based and Leave-One-Out (LOO) based. Shapley value based methods include Data Shapley (Ghorbani & Zou, 2019) and computationally feasible variations (Jia et al., 2020; Wang et al., 2024). LOO-based methods often encompass model retraining with one sample left out (Koh & Liang, 2020). In reality, rational participants in a market setting are likely to act strategically to maximize their profits. As a result, applying these pricing methods can lead to misreporting and inefficiencies within data markets.

**Data Pricing Methods.** Data pricing research spans several strands that formalize how data should be valued and traded. Early work in databases developed query-based pricing schemes that tie prices to the informational content of query results rather than entire datasets (Kushal et al., 2011; Koutris et al., 2015). Later models incorporated quality- and attribute-based considerations to reflect heterogeneity across data products Fricker & Maksimov (2017). A distinct direction introduces model-based pricing, where prices derive from the performance of machine-learning models trained on the data (Chen et al., 2019). Multi-party market models capture strategic interaction among data sellers and buyers through bargaining or cooperative-game formulations (Mi et al., 2025). Despite these advances, real data marketplaces still rely on coarse heuristics, leaving a persistent gap between theoretical pricing models and the context-dependent, non-rival utility that characterizes practical data exchange (Zhang et al., 2023).

**Auction Theory and Mechanism Design.** Starting from the Arrow–Debreu model (Arrow & Debreu, 1954), a central question of market design is to set prices such that the net welfare of all participants is maximized. Such a market is said to be *efficient*. Auction theory (Krishna, 2003), and in general mechanism design, investigates how to how to design prices which preserve efficiency even if some information is unknown i.e. *private*. In the context of data markets, this implies that we want to design prices such that all participants *truthfully* report their true costs of data sharing and benefits received from said data, so that we can maximize social welfare.

**Data Markets.** Due to unknown data sharing costs incurred by the data sellers, designing an efficient mechanism can be challenging. Dütting et al. (2021) show that with one data sample from the seller's distribution, truthful mechanisms can be achieved with approximate market efficiency, assuming unit supply sellers. Rasouli & Jordarn (2021) design a truthful mechanism where data quality can be exchanged with monetary payments. However, they do not analyze its social efficiency.

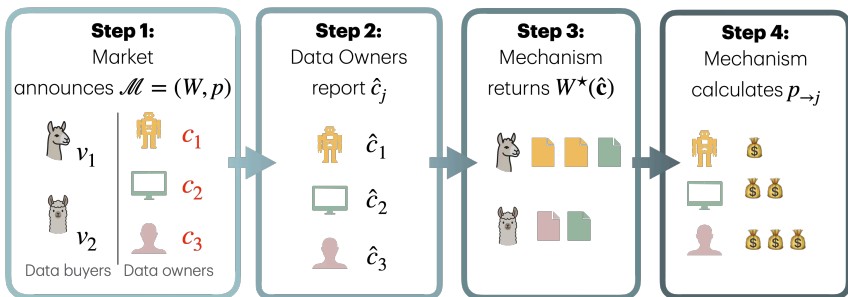

Figure 2: Diagram of our data market modeling framework. Red denotes private values.

Agarwal et al. (2024) study truthful mechanism design when buyers' externalities due to competition are known, aiming at maximizing social welfare or revenue. We refer to a recent survey (Zhang et al., 2024) for a more detailed overview of the area. We note that most of these works often assume known simple structured valuation functions (Agarwal et al., 2019) and combinatorial allocations (Moor et al., 2018; Colini-Baldeschi et al., 2020), not suitable for machine learning worlds where the information sharing can happen in continuous space and the buyers' valuations are directly connected to model performances, which is our focus.

## 3 WHAT IS A DESIRED PAYMENT?

### 3.1 OUR PROPOSED MODELING FRAMEWORK

We consider a data market with disjoint sets of data buyers $\mathcal{B}$ and data sellers $\mathcal{S}$. $\mathcal{B} \cap \mathcal{S} = \emptyset$. Buyers and sellers interact through the allocation (information exchange) matrix $\boldsymbol{W} \in \mathbb{R}^{|\mathcal{B}| \times |\mathcal{S}|}$, where $|w_{i,j}|$ denotes the information exchange between buyer $i$ and seller $j$, for example, how much data buyer $i$ purchases from seller $j$. Depending on the use case, $\boldsymbol{W}$ can live in constrained or unconstrained domains. For easy theoretical derivation, we further assume that $\boldsymbol{W}$ is a continuous variable. We use $\boldsymbol{W}_{i,:}$ and $\boldsymbol{W}_{:,j}$ to denote $i$-th row and $j$-th column, respectively. Our proposed modeling framework is illustrated in Figure 2.

**Buyer's Performance:** Each buyer has its performance (e.g. accuracy) function $v_i(\boldsymbol{W}_{i,:}) : \mathbb{R}^{|\mathcal{S}|} \to \mathbb{R}_{\geq 0}$. The performance measures how buyer $i$'s model gets improved utilizing the data from the sellers; or equivalently, the drop in loss after data acquisition, $v_i = l_i(\boldsymbol{0}) - l_i(\boldsymbol{W}_{i,:})$, where $l_i(\boldsymbol{0})$ denotes the standalone loss for buyer $i$. In game theory literature, such a performance function can as well be called buyer's valuation. In our main analysis, we assume buyer's valuation functions are known to the market, which makes the problem more tractable. It is infeasible to design a truthful and efficient market with unknown buyers' valuation, as analyzed in Appendix E.

**Seller's Cost:** $c_j \in \mathbb{R}_{\geq 0}$ is the *private unit cost* of data and $f(\cdot) : \mathbb{R}^{|\mathcal{B}|} \to \mathbb{R}_{\geq 0}$ quantifies the data sharing magnitude. For a specific seller $j$, we denote the data sharing magnitude as $f_j(\boldsymbol{W}) = f(\boldsymbol{W}_{:,j})$. $f_j(\boldsymbol{W})$ is non-decreasing in the norm of $j$th column of $\boldsymbol{W}$. A typical choice of $f_j$ is $\sum_{i \in \mathcal{B}} w_{ij}^2$. The *total* data sharing cost for seller $j$ is thus $c_j f_j(\boldsymbol{W})$. The notion of data sharing cost can encompass data generation costs, costs due to privacy leakage, and etc. Using game theory terminology, the seller's valuation is $-c_j f_j$.

**Assumption 1.** *All $v_i$ and $f_j$ are differentiable with respect to $\boldsymbol{W}$.*

**Definition 1** (Social Welfare and Social Cost). *For an allocation $\boldsymbol{W}$, social welfare (SW) is the sum of all players' valuations (buyers' performance minus sellers' costs): $SW = \sum_{i \in \mathcal{S}} v_i(\boldsymbol{W}_{i,:}) - \sum_{j \in \mathcal{B}} c_j f(\boldsymbol{W}_{:,j})$. Correspondingly, we can define social cost (SC), minimizing which is equivalent to maximizing the social welfare. $SC = \sum_{i \in \mathcal{S}} l_i(\boldsymbol{W}_{i,:}) + \sum_{j \in \mathcal{B}} c_j f(\boldsymbol{W}_{:,j})$*

**Mechanism.** A mechanism is a pair $\mathcal{M} = (\boldsymbol{W}, \boldsymbol{p})$, where $\boldsymbol{W}$ is the allocation matrix and $\boldsymbol{p} = (p_i)_{i \in \mathcal{B} \cup \mathcal{S}}$ is the vector of payments to/from the players. We consider several payment specifications in the sections that follow. Given reported costs $\hat{\boldsymbol{c}}$, the mechanism selects an allocation $\boldsymbol{W}^\star(\hat{\boldsymbol{c}})$ that maximizes reported social welfare:

$$\boldsymbol{W}^{\star}(\hat{\boldsymbol{c}}) \in \arg\max[SW := \sum_{i \in \mathcal{B}} v_i(\boldsymbol{W}) - \sum_{j \in \mathcal{S}} \hat{c}_j f_j(\boldsymbol{W})]. \tag{1}$$

An optimum $\boldsymbol{W}^{\star}(\hat{\boldsymbol{c}})$ always exists; see Appendix C.1.1. The mapping $\boldsymbol{W}^{\star}(\cdot)$ is fully determined by the reported cost $\hat{\boldsymbol{c}}$.

**Use Cases.** We list use cases where both buyers' performance function and sellers' data sharing costs can be modeled by $\boldsymbol{W}$.

1. Acquiring Data from Multiple Domains: When conducting pre-training runs for LLMs, it is usually needed to decide on the data mixtures from different domains/sellers. With a larger $w_{i,j}$, buyer $i$ benefits from more data from seller $j$, and for seller $j$, the costs for collecting and preparing the data also increase. $\boldsymbol{W} \in \{\boldsymbol{W} : w_{i,j} \geq 0 \text{ and } \sum_{j \in S} w_{i,j} = 1\}$.

2. Differential Private Model Sharing: When sharing models, seller $j$ shares a noisy version of the local model ($\boldsymbol{\theta}_j + \epsilon_{i,j}$) with buyer i, where $\epsilon_{i,j} \sim \mathcal{N}(0,1)$. As $|w_{i,j}|^2$ increases, less noise is added to the model shared by seller j, resulting in a higher degree of privacy. $\mathbf{W} \in \mathbb{R}^{N \times N}$.

3. Data Compensation in RAG Market: where $w_{ij} \in \{0,1\}$ denotes if document $j$ is retrieved for query $i$. Data contributor gets compensated once their data gets retrieved, and on the other hand, LLM generates better responses with retrieval augmentation.

## 3.2 Desired Properties for Payments

Desired payments should be truthful, individual-rational, budget-balanced and socially efficient. We formally define these properties as follows.

**Definition 2** (Social Efficiency). *Allocation denoted by $\boldsymbol{W} \in \mathcal{W}$ is socially efficient (SE) if it minimizes the social cost or maximizes the social welfare.*

**Definition 3** (Utility). *Let $P_{i\rightarrow}$ be the price to pay for a buyer $i \in \mathcal{B}$ and $P_{\rightarrow j}$ be the payment made to a seller $j \in \mathcal{S}$. Utility is defined as the participant's payoff from the market:*

$$u_i := v_i(\boldsymbol{W}) - P_{i\rightarrow}, \qquad u_j := P_{\rightarrow j} - c_j f_j(\boldsymbol{W})$$

**Definition 4** (Individual Rationality). *Individual Rationality (IR) is satisfied if each participant's utility is non-negative*

$$u_i \geq 0 \ \forall i \in \mathcal{B}, \qquad u_j \geq 0 \ \forall j \in \mathcal{S}.$$

**Definition 5** (Incentive Compatibility or Truthful). *A mechanism $\mathcal{M}$ is considered incentive-compatible (IC) if each participant achieves their best outcome by truthfully reporting their private values, regardless of what others report. We also refer to an IC mechanism as a* truthful *mechanism.*

**Definition 6** (Budget Balance). *A mechanism is called strong budget-balanced (SBB) if $\sum_{i \in \mathcal{B}} P_{i\rightarrow} = \sum_{j \in \mathcal{S}} P_{\rightarrow j}$ and weak budget-balanced (WBB) if $\sum_{i \in \mathcal{B}} P_{i\rightarrow} > \sum_{j \in \mathcal{S}} P_{\rightarrow j}$.*

## 4 Common Payments are not Truthful

In this section, we examine three common payment methods for seller compensation: direct reimbursement of reported costs and two widely used data valuation approaches, namely LOO and Data Shapley payments. We show that all of them incentivize sellers to misreport their unit data costs.

From now on, we denote $\hat{\boldsymbol{c}}_{-j} = \hat{\boldsymbol{c}} \backslash \{\hat{c}_j\}$, that is, the collection of all reported unit costs of sellers other than seller $j$. We first prove a fundamental result, which shows that with a larger reported unit data cost, seller $j$ shares less data.

**Claim 1.** $f_j(\boldsymbol{W}^{\star}(\hat{c}_j, \hat{\boldsymbol{c}}_{-j}))$ *is monotonically non-increasing in $\hat{c}_j$.*

Proof can be seen in Appendix C.2.1. It is worth noting that the total data sharing magnitude of other sellers than $j$ (denoted by $\sum_{k \in \mathcal{S} \backslash \{j\}} f_k(\boldsymbol{W}^{\star}(\hat{c}_j, \hat{\boldsymbol{c}}_{-j}))$) can increase or decrease in $\hat{c}_j$, as $\boldsymbol{W}^{\star}(\hat{c}_j, \hat{\boldsymbol{c}}_{-j}))$ can have both positive and negative entries. We illustrate this empirically in Figure 8.

## 4.1 Direct Reimbursement Incentivizes Over-reporting

The most straightforward payment is to directly pay data sellers the costs they report, denoted as $P_{\to j}^{\text{dir}} = \hat{c}_j f_j(\boldsymbol{W}^\star(\hat{c}_j, \hat{\boldsymbol{c}}_{-j}))$. We show that such a payment incentivizes overreporting $\hat{c}_j$, i.e. $\hat{c}_j > c_j$.

**Claim 2.** *Using reported cost as naive payment incentivizes data owners to over-report $\hat{c}_j$.*

*Proof.* The utility for seller $j$ is the payment it received minus its data sharing cost.

$$u_j = P_{\to j}^{\text{dir}} - c_j f_j(\boldsymbol{W}^\star(\hat{c}_j, \hat{\boldsymbol{c}}_{-j})) = (\hat{c}_j - c_j) f_j(\boldsymbol{W}^\star(\hat{c}_j, \hat{\boldsymbol{c}}_{-j})) \quad (2)$$

When $\hat{c}_j > c_j$, we have $u_j(\hat{c}_j) > u_j(c_j) = 0$. Strategic seller $j$ would over-report. $\square$

Notably, the seller's incentive to over-report is bounded as $f_j(\boldsymbol{W}^\star(\hat{c}_j, \hat{\boldsymbol{c}}_{-j}))$ approaches zero when $\hat{c}_j$ becomes sufficiently large, resulting in near-zero utility. This is observed in Figure 3.

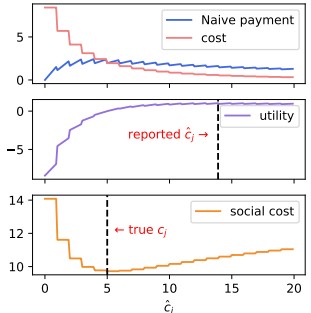

Figure 3: User's utility is maximized when reported $\hat{c}_j$ is greater than true $c_j = 5$ (Mean estimation market).

### 4.2 POPULAR DATA VALUATION PAYMENTS ARE NOT TRUTHFUL EITHER

We now review two conventional data valuation methods: Leave-One-Out (LOO) valuation and Data Shapley. When seller compensation is based on these methods, we demonstrate that sellers are consistently incentivized to misreport their private types, as compensation depends on their self-reported information.

#### 4.2.1 LEAVE-ONE-OUT PAYMENT RULE

LOO has been a standard approach to estimate influence in statistics (Weisberg & Cook, 1982; Koh & Liang, 2020). LOO valuation quantifies a data point's contribution by assessing the change in model performance when the point is removed. This method inherently requires retraining the model on the reduced dataset. In our context, it corresponds to the following equation, representing the difference in buyers' performance function with and without the presence of data seller $j$.

$$P_{\to j}^{LOO} = \sum_{i \in \mathcal{B}} v_i(\boldsymbol{W}^\star(\hat{c}_j, \hat{\boldsymbol{c}}_{-j})) - v_i(\boldsymbol{W}^\star(\infty, \hat{\boldsymbol{c}}_{-j})) \quad (3)$$

**Claim 3.** *Using LOO valuation as the payment rule incentivizes sellers to misreport their true costs, either by over-reporting or under-reporting (proof in Appendix C.2.3).*

#### 4.2.2 PAYMENT VIA SHAPLEY VALUE

Shapley value was first introduced to attribute fair contributions in a cooperative game by Shapley (1951). In the Machine Learning world, Data Shapley (Ghorbani & Zou, 2019) was proposed to address data valuation. We adopt data shapley calculation in our scenario, as in (4). The idea is to calculate the marginal contribution of seller $j$, averaged over *all* subsets of sellers. Depending on the market size, the calculation of Shapley value can be computationally inefficient. In contrast, LOO payment only considers the marginal contribution to the subset $\mathcal{S} \backslash \{j\}$.

$$P_{\to j}^{SP} = \sum_{\pi \subseteq \mathcal{S} \backslash \{j\}} \frac{|\pi|! \left(|\mathcal{S}| - |\pi| - 1\right)!}{|\mathcal{S}|!} \left[ \sum_{i \in \mathcal{B}} v_i\big(\mathcal{B} \cup \pi \cup \{j\}\big) - \sum_{i \in \mathcal{B}} v_i\big(\mathcal{B} \cup \pi\big) \right]. \quad (4)$$

where $v_i\big(\mathcal{B} \cup \pi\big)$ denotes $v_i(\boldsymbol{W}')$, where $\boldsymbol{W}'$ maximizes the reported social welfare of buyers $\mathcal{B}$ and a subset of sellers $\pi$: $\boldsymbol{W}' = \arg\max_{\boldsymbol{W} \in \mathbb{R}^{|\mathcal{B}| \times |\pi|}} \sum_{i \in \mathcal{B}} v_i(\boldsymbol{W}) - \sum_{j \in \pi} \hat{c}_j f(\boldsymbol{W})$.

**Remark 1.** *$P_{\to j}^{SP}$ is not IC either. As like LOO payment, the utility of seller $j$ is dependent on its reported $\hat{c}_j$, and thus a rational seller $j$ can manipulate $\hat{c}_j$ to arrive at a higher utility.*

**Remark 2.** *If the performance function is super-additive[1] (e.g. when complementary but necessary data sources are combined to solve a task), LOO payment is greater than Shapley payment. On the*

---

[1] $v_i$ specified by $\boldsymbol{W}^\star(\hat{\boldsymbol{c}})$ are super-additive in the set of data sellers

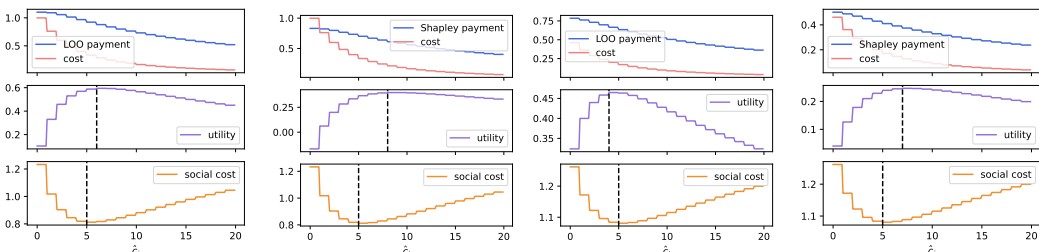

Figure 4: First row: how $P_{\to j}^{LOO}$ and $P_{\to j}^{SP}$ change with reported unit cost $\hat{c}_j$; second row: how utility ($u_j = P_{\to j} - c_j f_j(\boldsymbol{W}^\star(\hat{c}_j, \hat{\boldsymbol{c}}_{-j}))$) change with $\hat{c}_j$; third row: how social cost ($SC = \sum_{i\in\mathcal{B}} v_i(\boldsymbol{W}^\star(\hat{c}_j, \hat{\boldsymbol{c}}_{-j})) - \sum_{k\in\mathcal{S}} c_k f_k(\boldsymbol{W}^\star(\hat{c}_j, \hat{c}_{-j}))$) change with $\hat{c}_j$. The dashed lines denote optima. The first and second pictures are from one random seed, and the third and fourth pictures are from another random seed in our Mean Estimation Market experiment.

*flip side, if the performance function is sub-additive[2] (e.g. when the contribution of one user is very similar to that of another), LOO payment is smaller than Shapley payment.*

### 4.3 AN ILLUSTRATIVE EXAMPLE – MEAN ESTIMATION MARKETS

To support our claim that LOO and Shapley payments can incentivize misreporting, we present a simple illustrative example based on mean estimation tasks. In this setup, buyers aim to estimate their local means $\{\boldsymbol{\mu}_i\}_{i\in\mathcal{B}}$. Sellers have *unbiased* local estimates from their local samples sampled from $\mathcal{N}(\boldsymbol{\mu}_j, \sigma_j^2)$, $j \in \mathcal{S}$. In order to achieve lower MSE loss, buyers may seek trading with sellers for their local estimates and combine them with weights $\boldsymbol{W}$. We choose $f_j$ to be $\sum_{i\in\mathcal{B}} w_{ij}^2$. $\boldsymbol{W}$ can be calculated by minimizing social cost, which is the sum of MSE and data sharing costs.

$$\boldsymbol{W}^\star(\hat{c}) \in \arg\min \left[ SC \coloneqq \mathbb{E}\sum_{i\in\mathcal{B}} \|\sum_{j\in\mathcal{S}} w_{i,j}\hat{\boldsymbol{\mu}}_j - \boldsymbol{\mu}_i\|^2 + \sum_{j\in\mathcal{S}} \hat{c}_j \sum_{i\in\mathcal{B}} w_{i,j}^2 \right] \tag{5}$$

Due to the unique quadratic structure of the objective, we have a closed-form solution of $\boldsymbol{W}$ in (6).
$$\boldsymbol{W}^\star(\hat{c}) = \boldsymbol{B}(\boldsymbol{C} + \boldsymbol{V} + \boldsymbol{A})^{-1} \tag{6}$$

where $\boldsymbol{B} = [\langle \boldsymbol{\mu}_i, \boldsymbol{\mu}_j \rangle]_{i\in\mathcal{B}, j\in\mathcal{S}}$, $\boldsymbol{C} = [\langle \boldsymbol{\mu}_i, \boldsymbol{\mu}_k \rangle]_{i,j\in\mathcal{S}}$, $\boldsymbol{V} = diag(\sigma_j^2)$ and $\boldsymbol{A} = diag(\hat{c}_j)$. The proof is provided in Appendix C.2.2.

Thanks to the trackable formulation, we can thus calculate different payments precisely and efficiently. From our simulations detailed in Appendix D.1.1, we are able to accurately track how SC, utilities and payments change with $\hat{c}_j$ in Figure 4. It is clear that when both LOO and Data Shapley are used as payment methods, the data owners are incentivized to misreport to maximize their own utilities, which increases the social cost for the whole marketplace.

## 5 GAME-THEORETIC TRUTHFUL PAYMENT RULES

Since popular data valuation strategies perform poorly with respect to truthfulness, we turn to established game-theoretical results that guarantee truthful *seller* reports. We adapt Myerson payment and VCG payment in our data market scenario and characterize their properties. We further demonstrate a scenario and a payment distribution rule where buyer IR is satisfied.

### 5.1 MYERSON PAYMENT RULE

Myerson payment is a classical truthful payment rule (Myerson, 1981), which pays sellers the reported cost plus an integral term. Our specific scenario maps to the following equation in (7). Myerson payment rule is IC and IR for all sellers, for which we provide a proof in Appendix C.3.1.

$$P_{\to j}^{MS} = \hat{c}_j f_j(\boldsymbol{W}^\star(\hat{c}_j, \hat{\boldsymbol{c}}_{-j})) + \int_{\hat{c}_j}^\infty f_j(\boldsymbol{W}^\star(u, \hat{\boldsymbol{c}}_{-j}))du \tag{7}$$

---

[2] $v_i$ specified by $\boldsymbol{W}^\star(\hat{\boldsymbol{c}})$ are sub-additive in the set of data sellers

## 5.2 VCG PAYMENT RULE

VCG payment (Vickrey, 1961; Clarke, 1971; Groves, 1973) is another classical payment rule, which calculates the externality of a specific seller to the social welfare, as in (8). The idea is to calculate the differences between the social welfare of $\mathcal{B} \cup \mathcal{S} \setminus \{j\}$ in seller $j$'s absence and the welfare when seller $j$ is present. By design, VCG payment rule is IC and IR for all sellers, for which we provide a proof in Appendix C.3.1. Moreover, VCG payment is upper-bounded.

$$
P_{\rightarrow j}^{VCG} = \sum_{i \in \mathcal{B}} v_i(\boldsymbol{W}^\star(\hat{c}_j, \hat{\boldsymbol{c}}_{-j})) - \sum_{k \in \mathcal{S} \setminus \{j\}} \hat{c}_k f_k(\boldsymbol{W}^\star(\hat{c}_j, \hat{\boldsymbol{c}}_{-j}))
$$
$$
- \left[ \sum_{i \in \mathcal{B}} v_i(\boldsymbol{W}^\star(\infty, \hat{\boldsymbol{c}}_{-j})) - \sum_{k \in \mathcal{S} \setminus \{j\}} \hat{c}_k f_k(\boldsymbol{W}^\star(\infty, \hat{\boldsymbol{c}}_{-j})) \right]
\tag{8}
$$

**Claim 4.** $P_{\rightarrow j}^{VCG} \leq \sum_{i \in \mathcal{B}} v_i(\boldsymbol{W}^\star(c_j, \boldsymbol{c}_{-j})) - v_i(\boldsymbol{W}^{\star - j}(c_j, \boldsymbol{c}_{-j}))$, where $\boldsymbol{W}^{\star - j}$ is $\boldsymbol{W}^\star$ with the $j$th column set to zero, and all other entries unchanged. (Proof in Appendix C.4)

## 5.3 CHARACTERIZATION OF THE TRUTHFUL PAYMENT RULES

Myerson and VCG payment rules ensure both IC and IR for data sellers. How do they compare to each other? Theorem 5.1 shows that in general, Myerson is the smallest payment rule, thus optimal from the buyers' perspective. In certain scenarios, i.e. when $\boldsymbol{W}$ lives in an unconstrained domain, we further have VCG payment equivalent to Myerson payment, which is proven in Theorem 5.2.

**Theorem 5.1.** *Myerson payment is the smallest IC and IR (for data sellers) payment rule.*

Proof can be checked at Appendix C.4.1.

**Theorem 5.2.** *Myerson payment is equivalent to the VCG payment when the domain of $\boldsymbol{W}$ is unconstrained.*

*Proof sketch.* As $\boldsymbol{W}^\star$ is chosen to maximize the reported SW, we have $\frac{\partial v_i(\boldsymbol{W}_{i,:}^\star)}{\partial w_{i,j}^\star} - \hat{c}_j \frac{\partial f(\boldsymbol{W}_{:,j}^\star)}{\partial w_{i,j}^\star} = 0$, for all $w_{i,j}$. Following this, we can show that $\frac{\partial SW(\boldsymbol{W}^\star(\hat{c}_j, \hat{\boldsymbol{c}}_{-j}))}{\partial \hat{c}_j} = -f_j$. Plugging this into the integral calculation of Myerson payment, we prove the claim. Full proof in App C.4.2. □

**Discussion.** Myerson payment rule provides theoretical guarantees of being buyer-optimal. However, its computation can be costly since $f_j(\boldsymbol{W}^\star(u, \hat{\boldsymbol{c}}_{-j}))$ requires optimizing $SW(u, \hat{\boldsymbol{c}}_{-j})$, the complexity of which depends on the structure of $v_i$ and $f_j$. VCG payment, on the other hand, is more computationally feasible. Yet, it still requires leave-one-out retraining. Future work should explore more computationally efficient implementations.

## 5.4 CAN WE ENSURE IR FOR DATA BUYERS?

Until now, we have focused solely on the seller side. But can our mechanism also benefit data buyers? Interestingly, we can redistribute the payments made to sellers among buyers based on their marginal contributions. This ensures SBB in the market and guarantees IR for data buyers when the performance function is subadditive.

**Theorem 5.3.** *Assume $v_i$ is subadditive (i.e. has diminishing returns in $\boldsymbol{W}$). A payment rule defined as follows is IR for all buyers.*

$$
\eta_{ij} = \frac{v_i(\boldsymbol{W}^\star(\boldsymbol{c})) - v_i(\boldsymbol{W}^{\star - j}(\boldsymbol{c}))}{\sum_{k \in \mathcal{B}} v_k(\boldsymbol{W}^\star(\boldsymbol{c})) - v_k(\boldsymbol{W}^{\star - j}(\boldsymbol{c}))}, \qquad P_{i \rightarrow j} = \eta_{ij} P_{\rightarrow j}
\tag{9}
$$

*Proof sketch.* From Theorem 5.1, we know that the Myerson payment is always less than or equal to the VCG payment. In Claim 4, we further upper bound the VCG payment by $\overline{P}_{\rightarrow j}$. It is straightforward to verify that distributing $\overline{P}_{\rightarrow j}$ among the data buyers according to the distribution rule in Equation (9) ensures IR for all participants. Any smaller payment would also satisfy the IR condition by construction. A detailed proof is provided in Appendix C.4.3. □

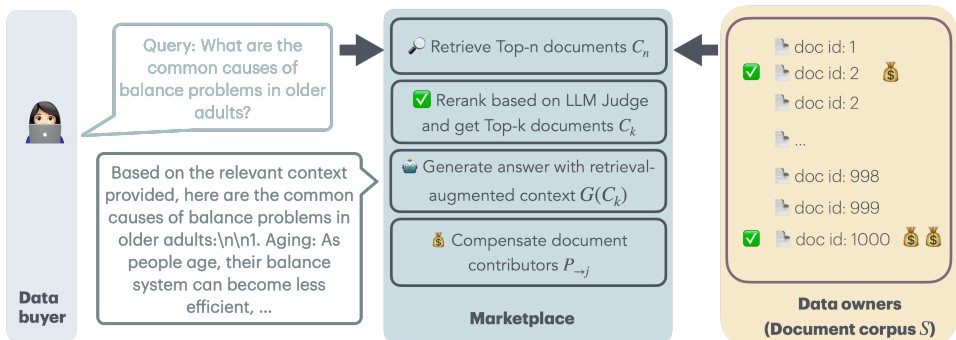

Figure 5: A example of our designed RAG marketplace. Data owners supply proprietary curated datasets to improve question answering in exchange for compensation. Each of the curated dataset varies in relevance to the buyer, quality of the dataset itself, as well as the cost to the data owner. Experimental details are presented in Appendix D.3.

**Remark 3.** *When the performance function $v_i$ is super-additive, we currently lack a solution that guarantees individual rationality (IR) for the buyer. Whether it is possible to design a suitable buyer payment mechanism remains an open question.*

## 6 EXPERIMENTS WITH RAG MARKET

In this section, we show how our data-market framework applies to a realistic setting: compensating content contributors in a RAG market. Unlike the earlier setup with continuous allocation, the RAG setting features discrete allocation. The core takeaway is unchanged: Data Shapley and LOO do not generally ensure truthfulness, whereas Myerson and VCG payments do.

### 6.1 WHAT IS RAG?

RAG is short for Retrieval-Augmented Generation (Lewis et al., 2021). Before generating an answer, it retrieves relevant documents from a knowledge source, then provides these documents as context for LLMs to produce more accurate and grounded responses. It consists of two main parts: a retriever $R$ and a generator $G$. $R$ takes in the user query $q$ and a document corpus of $s$ documents $\mathcal{S}$ (i.e. $|\mathcal{S}| = s$), and returns $R(q, \mathcal{S}) \to \mathbb{R}^s$ the set of all relevance scores. Top $n$ documents are picked based on the relevance score, resulting $\mathcal{C}_n \subset \mathcal{S}$. Usually there is a post-retrieval process to rerank the documents (typically via an LLM judge) to avoid information overload (Gao et al., 2024). After reranking, a subset of $k$ documents $\mathcal{C}_k \subset \mathcal{S}$ is passed to the generator $G$ to generate output $G(\mathcal{C}_k)$.

In practice, there is a max $k \leq n$ limit on the documents that RAGs can utilize. This may be because of constraints on the context-window (can fit at most $k$ documents) or on the inference latency (latency typically scales quadratically with context length). Hence, we limit the selection $\mathcal{C}$ to size $k$.

### 6.2 RAG MARKETPLACE SETUP

In this marketplace, we want to retrieve relevant documents according to user queries at low costs, and at the same time, incentivize document owners to report true data costs. Contrast to the previous analysis, here allocation $\boldsymbol{w} \in \{0, 1\}^s$ is discrete, with each entry indicating whether a document is utilized at the end, i.e., after reranking. In the RAG setting, we consider a single-buyer scenario, since in practice users are charged per query (i.e., per purchase). Consequently, $\boldsymbol{w}$ is a vector rather than a matrix. We choose $f_j(\boldsymbol{w}) = w_j^2$, that is $f_j = w_j \in \{0, 1\}$.

**Valuation.** For a selected subset $\mathcal{C} \subseteq \mathcal{S}$ where $\mathcal{C} = \{\text{document}_i \mid w_i = 1\}_{i \in [s]}$. The resulting valuation of a selection $\boldsymbol{w} \in \{0, 1\}^s$ is defined as the LLM Judge's evaluation score over the response $G(\mathcal{C})$, on how the context-enhanced response answers the query $q$.

$$v(\boldsymbol{w}) = \text{LLM-Judge}(q, G(\mathcal{C})) \in [0, 10] \text{ evaluating the response } G(\mathcal{C}) \text{ on the query } q. \quad (10)$$

**Selection/allocation.** The standard RAG process does not take costs into account. To address this, we modify the reranking process. When reranking, we prioritize documents with low costs. The

allocation is thus a result of similarity-based retrieval and cost-aware reranking in (11). Given the reported costs $\hat{c}_1, \ldots \hat{c}_s$, we find the allocation $\boldsymbol{w}$ that maximizes the social welfare i.e.

$$\boldsymbol{w}^* := \arg\max_{\boldsymbol{w}} \left[ SW = v(\boldsymbol{w}) - \hat{\boldsymbol{c}}^\top \boldsymbol{w} \right], \text{ or equivalently} \tag{11}$$

$$\{w_i^* := 1 \text{ if } \text{doc}_i \in \mathcal{C}^* \text{ o. w. } 0\}_{i \in [s]}, \text{ where}$$

$$\mathcal{C}^* := \arg\max_{\mathcal{C}' \subseteq \mathcal{C}_n \text{ s.t. } |\mathcal{C}'| \le k} \text{LLM-Judge}(q, G(\mathcal{C}')) - \sum_{\{i: \text{doc}_i \in \mathcal{C}'\}} \hat{c}_i \tag{12}$$

**Payment.** We only pay a document provider if it gets picked in the $\mathcal{C}_n$ i.e. it has sufficiently high similarity to the query as judged by the retriever. The exact payment is determined by the marketplace and the specific mechanism as shown in Figure 5. Note that all formulas from previous sections defined in the continuous domain still apply, apart from Myerson payment. In this specific case, $f_j(\cdot)$ is a step function that jumps from 1 to 0 when the reported unit cost is over a threshold $\bar{c}_j$. Thus, Myerson payment turns into $P_{\to j}^{\text{MS}} = c_j + (\bar{c}_j - c_j)$. $\bar{c}_j$ is the largest unit cost seller $j$ can report beyond that it will no longer gets retrieved by the reranker. We present pseudocode for an instance of payment calculation in Appendix D.4.

### 6.3 Comparing and Evaluating Payment Mechanisms

**Unique dynamics in the discrete case.** Unlike the continuous case, where LOO and Shapley payments vary with the reported cost, in the discrete RAG setting, these payments are independent of the reported cost. This is because LOO/Shapley payments are dependent on the buyer valuation $v(\boldsymbol{w}(\hat{\boldsymbol{c}}))$. Since each entry of $\boldsymbol{w}$ can either be 0 or 1, once a document is retrieved ($w_j = 1$), buyer's valuation won't change with $\hat{c}_j$. However, the retrieval decision itself still depends on the reported cost. Since Shapley/LOO payments have no IR guarantees, it can be the case that Shapley/LOO payments cannot cover the data costs when the data unit costs are relatively large. In this scenario, data sellers will over-report to not be picked by the reranker, resulting in retrieval failures.

**Real-life validation.** We carry out a real-world experiment in a challenging medical-domain chatbot setting. A user can pose a medical question, and the language model can search among the provided medical knowledge base and offer retrieval grounded answers. The knowledge base is constructed using MedQuAD dataset (Ben Abacha & Demner-Fushman, 2019). We use DeepSeek-R1 (DeepSeek-AI, 2025) to grade[3] the retrieval-augmented answers by Qwen2.5-3B model (Yang et al., 2024). The experimental results are presented in Figure 6, which confirms the above dynamics. Furthermore, under the LOO, VCG, and Myerson payment mechanisms, only the retrieved documents in the final set $\mathcal{C}_k$ receive compensation. In contrast, the Shapley method allocates payment to all documents in $\mathcal{C}_n$, offering a more equitable approach for content contributors. As expected, Myerson and VCG mechanisms maintain truthfulness and individual rationality in all cases. In this specific scenario, we further have Myerson payment equivalent to VCG payment. The same analysis applies to multi-document retrieval, as illustrated by an example in Appendix D.5.

**Largely reduced computational complexity in the discrete case.** Because $\boldsymbol{w}$ takes discrete values, computing the Myerson and VCG payments becomes inexpensive, making the approach more practical for real-world deployment. Concretely, the integral in Myerson payment corresponds to the area under the curve of $f_j(\cdot)$ from $c_j$ to $\infty$. In the discrete setting, $f_j$ reduces to a step function that drops from 1 to 0 once document $j$ is no longer selected by (11). For VCG payment, the extra complexity lies in one more time of generation and retrieval, which is rather fast as well.

### 7 Conclusion

We revisit data valuation through the lens of market design and highlight the shortcomings of commonly used methods like Leave-One-Out and Data Shapley when applied as pricing rules. We demonstrate that these approaches can incentivize strategic misreporting, undermining market efficiency. By adapting classical mechanisms such as Myerson and VCG to the data market context, we provide truthful, individually rational, and socially efficient alternatives. Our framework not only offers theoretical guarantees but is also practical, as shown in applications like RAG market.

---

[3]We validate the LLM judge's scoring quality in Appendix D.3.3

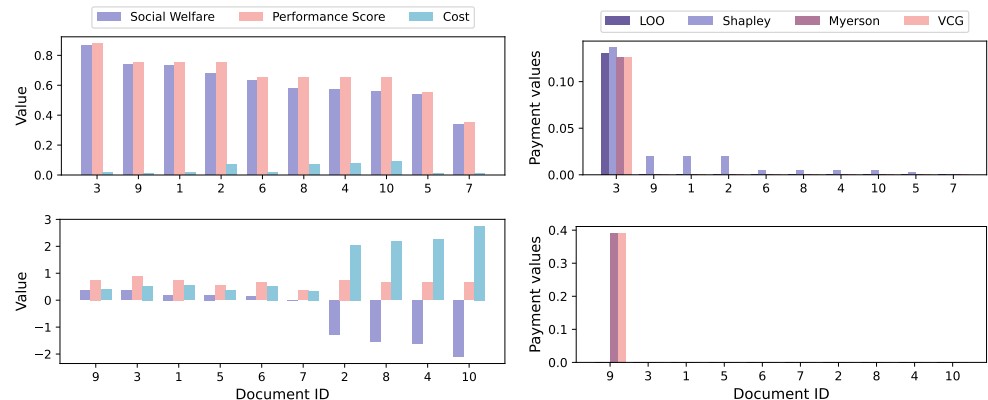

Figure 6: RAG market with $(n, k) = (10, 1)$ (document IDs are ranked according to the corresponding social welfare when the respective document is retrieved). The documents' unit costs are randomly generated. Upper row: low unit costs $(\hat{c})$; bottom row: high unit costs. With higher unit costs, LOO and Shapley payments fail to cover the data costs, incentivizing document owners to overreport; as a result, no documents are retrieved or compensated.

Our findings open several directions for future work: 1) how to make the existing truthful payment rule more computationally efficient? 2) how to improve the truthfulness of existing data valuation methods? 3) how to handle other strategic behaviors by sellers like adversarial data?, and 4) designing McAffee's trade reduction style mechanisms (McAfee, 1992) when both buyers and sellers have private valuations. More generally, a critical question facing research on data-market design is to identify the exact inefficiencies in the marketplace that prices could alleviate. We point out that when data costs are private and heterogeneous, prices can be used to truthfully surface these costs. There are likely other roles for prices as well. Taken together, we hope our work initiates an important and fruitful line of research on designing practical pricing strategies with game-theoretic considerations.

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

# A LIMITATIONS

Our analysis assumes a one-sided market where buyers' valuation functions are known—a simplification that enables theoretical traceability, but is rather unrealistic. We also consider the more general two-sided setting where buyers' valuations are private; however, only impossibility results can be established in this case.

Additionally, our theoretical results are derived under the assumption of a continuous allocation domain for $\boldsymbol{W}$. In the discrete setting, such as in our RAG market example, we provide only empirical findings. We encourage future work to extend these results and develop a more comprehensive understanding across both settings.

# B STATEMENTS

## B.1 ETHICS STATEMENT

We target an emerging scenario, as data is becoming more important than ever. Our work has a positive societal impact by offering a mechanism to compensate data contributors, potentially mitigating copyright violations. Furthermore, the proposed payment scheme discourages strategic misreporting by sellers, fostering a healthier and more efficient data marketplace.

## B.2 LLM USAGE STATEMENT

We used LLMs to polish writing as well as modifying plotting scripts. Furthermore, we conducted experiments using LLMs as generators and judges in the RAG market. We detailed the usage in Appendix D.3.

# C MISSING PROOFS

## C.1 SECTION 3 PROOFS

### C.1.1 PROOF FOR THE EXISTENCE OF OPTIMAL ALLOCATION

To guarantee the feasibility of the problem, we need to ensure that $\boldsymbol{W}^\star$ can be obtained as optimum for

$$max[g(\boldsymbol{W}) \coloneqq \sum_{i \in \mathcal{B}} v_i(\boldsymbol{W}) - \sum_{j \in \mathcal{S}} \hat{c}_j f_j(\boldsymbol{W})]$$

Apart from the continuity assumption, we further have $v_i$ upper bounded by $l_i(0)$ and $f_j$ bounded from below (by definition, we have $f_j \geq 0$), which stops the objective $g(\boldsymbol{W})$ from exploding upward.

Now, it is left to show that the optimum is attainable. According to Claim 1, $f_j$ is non-decreasing in the norm of j-th column of $\boldsymbol{W}$, denoted as $\boldsymbol{W}_{:,j}$, we have $\lim_{\boldsymbol{W} \to \infty} g(\boldsymbol{W}) \to -\infty$. Thus, we have the argmax $\boldsymbol{W}^\star < \infty$, that is, $\boldsymbol{W}^\star$ is feasible.

## C.2   SECTION 4 PROOFS

### C.2.1   PROOF FOR CLAIM 1

$W^\star(\hat{c})$ is chosen to optimize the following function according to our setup

$$\max \sum_{i \in \mathcal{B}} v_i(W^\star(\hat{c})) - \sum_{j \in \mathcal{S}} \hat{c}_j f_j(W^\star(\hat{c}))$$

We prove by contradiction. Suppose there exist two reported costs $c_j < c_j'$ such that:

$$f_j(W^\star(c_j', c_{-j})) > f_j(W^\star(c_j, c_{-j}))$$

By optimality of the allocation $W$, we have that:

$$\sum_{i \in \mathcal{B}} v_i(W^\star(c_j, c_{-j})) - c_j f_j(W^\star(c_j, c_{-j})) - \sum_{k \neq j} c_k f_k(W^\star(c_j, c_{-j}))$$
$$\geq \sum_{i \in \mathcal{B}} v_i(W^\star(c_j', c_{-j})) - c_j f_j(W^\star(c_j', c_{-j})) - \sum_{k \neq j} c_k f_k(W^\star(c_j', c_{-j})) \tag{13}$$

and similarly,

$$\sum_{i \in \mathcal{B}} v_i(W^\star(c_j', c_{-j})) - c_j' f_j(W^\star(c_j', c_{-j})) - \sum_{k \neq j} c_k f_k(W^\star(c_j', c_{-j}))$$
$$\geq \sum_{i \in \mathcal{B}} v_i(W^\star(c_j, c_{-j})) - c_j' f_j(W^\star(c_j, c_{-j})) - \sum_{k \neq j} c_k f_k(W^\star(c_j, c_{-j})). \tag{14}$$

Adding these two inequalities together, we find:

$$(c_j' - c_j) \left[ f_j(W^\star(c_j', c_{-j})) - f_j(W^\star(c_j, c_{-j})) \right] \leq 0.$$

Since by assumption $c_j' > c_j$, it must be the case that:

$$f_j(W^\star(c_j', c_{-j})) \leq f_j(W^\star(c_j, c_{-j})),$$

contradicting our initial assumption. Therefore, the function $f_j$ is monotonically non-increasing in the seller's reported cost $c_j$

### C.2.2   DERIVATION OF W IN MEAN ESTIMATION MARKET

We offer the proof for the derivation of closed-form $W$ in Section 4.3. The construction of mean estimation market is inspired by Theorem 2 from Lee et al. (2022).

$$SC := \mathbb{E} \sum_{i \in \mathcal{B}} \| \sum_{j \in \mathcal{S}} w_{i,j} \hat{\mu}_j - \mu_i \|^2 + \sum_{j \in \mathcal{S}} \hat{c}_j \sum_{i \in \mathcal{B}} w_{i,j}^2 \tag{15}$$

$$\mathbb{E} \| \sum_{j \in \mathcal{S}} w_{i,j} \hat{\mu}_j - \sum_{j \in \mathcal{S}} w_{i,j} \mu_j + \sum_{j \in \mathcal{S}} w_{i,j} \mu_j - \mu_i \|^2 + \sum_{j \in \mathcal{S}} \hat{c}_j w_{i,j}^2$$
$$= \sum_{j \in \mathcal{S}} w_{i,j}^2 \sigma_j^2 + (\sum_{j \in \mathcal{S}} w_{i,j} \mu_j)^\top (\sum_{j \in \mathcal{S}} w_{i,j} \mu_j)$$
$$- 2 \sum_{j \in \mathcal{S}} w_{i,j} \mu_j^\top \mu_i + \mu_i^\top \mu_i + \sum_{j \in \mathcal{S}} \hat{c}_j w_{i,j}^2 \tag{16}$$
$$= W_{i,:}(V + A) W_{i,:}^\top + W_{i,:} C W_{i,:}^\top - 2 W_{i,:} B_{i,:}^\top + \mu_i^\top \mu_i$$

Let $B = [\langle \mu_i, \mu_j \rangle]_{i \in \mathcal{B}, j \in \mathcal{S}}$, $C = [\langle \mu_i, \mu_k \rangle]_{i,j \in \mathcal{S}}$, $V = diag(\sigma_j^2)$ and $A = diag(\hat{c}_j)$. We have

$$W_{i,:}^\top = (V + A + C)^{-1} B_{i,:}^\top$$

Thus,

$$\boldsymbol{W}^{\star}(\hat{c}) = \boldsymbol{B}(\boldsymbol{V} + \boldsymbol{A} + \boldsymbol{C})^{-1}$$

$$SC = \sum_{i \in \mathcal{B}} \left[ \boldsymbol{\mu}_i^{\top} \boldsymbol{\mu}_i - \boldsymbol{B}_{i,:}(\boldsymbol{V} + \boldsymbol{A} + \boldsymbol{C})^{-1} \boldsymbol{B}_{i,:}^{\top} \right]$$

### C.2.3 Proof for Claim 3

*Proof.* For seller $j$, the utility is

$$
\begin{aligned}
u_j &= P_{\to j}^{LOO} - c_j f(\boldsymbol{W}^{\star}(\hat{c}_j, \hat{\boldsymbol{c}}_{-j})) \\
&= \sum_{i \in \mathcal{B}} v_i(\boldsymbol{W}^{\star}(\hat{c}_j, \hat{\boldsymbol{c}}_{-j})) - v_i(\boldsymbol{W}^{\star}(\infty, \hat{\boldsymbol{c}}_{-j})) - c_j f(\boldsymbol{W}^{\star}(\hat{c}_j, \hat{\boldsymbol{c}}_{-j}))
\end{aligned}
\tag{17}
$$

Let the constrained set be

$$\Omega = \left\{ \mathbb{R}^{|\mathcal{B}| \times |\mathcal{S}|} \ \middle| \ \psi_q(\boldsymbol{W}) \le 0 \ \text{ for every } q \right\},$$

where $q$ is an index labeling the different constraint functions imposed on the allocation matrix $\boldsymbol{W}$.

Let $\Phi(\boldsymbol{W}^{\star}(\mathbf{c}))$ denote the social welfare when the costs are $\mathbf{c} \in \mathbb{R}^{|\mathcal{S}|}$. The stationarity condition from the Karush–Kuhn–Tucker (KKT) conditions is

$$\nabla_{\boldsymbol{W}} \Phi(\boldsymbol{W}^{\star}(\hat{\mathbf{c}})) + \sum_q \lambda_q \nabla_{\boldsymbol{W}} \psi_q(\boldsymbol{W}^{\star}(\hat{\mathbf{c}})) = 0,$$

together with complementary slackness

$$\lambda_q \psi_q(\boldsymbol{W}) = 0, \quad \lambda_q \ge 0.$$

We now check

$$\frac{d}{d\hat{c}_j} \Phi(\boldsymbol{W}^{\star}(\hat{\mathbf{c}})) = \nabla_{\boldsymbol{W}} \Phi(\boldsymbol{W}^{\star}(\hat{\mathbf{c}})) \cdot \frac{d\boldsymbol{W}^{\star}(\hat{\mathbf{c}})}{d\hat{c}_j} + \frac{\partial \Phi(\boldsymbol{W}^{\star}(\hat{\mathbf{c}}))}{\partial \hat{c}_j}.$$

The first term vanishes by the KKT condition, while the second term equals $-f_j(\boldsymbol{W}^{\star}(\hat{\mathbf{c}}))$. Thus,

$$\frac{d}{d\hat{c}_j} \Phi(\boldsymbol{W}^{\star}(\hat{\mathbf{c}})) = -f_j(\boldsymbol{W}^{\star}(\hat{\mathbf{c}})).$$

Next, consider the utility of seller $j$:

$$u_j = P_{\to j}^{\text{LOO}} - c_j f_j(\boldsymbol{W}^{\star}(\hat{\mathbf{c}})).$$

By expanding $P_{\to j}^{\text{LOO}}$, we can write

$$u_j = \Phi(\boldsymbol{W}^{\star}(\hat{\mathbf{c}})) + (\hat{c}_j - c_j) f_j(\boldsymbol{W}^{\star}(\hat{\mathbf{c}})) + \sum_{k \ne j} \hat{c}_k f_k(\boldsymbol{W}^{\star}(\hat{\mathbf{c}})) + h_{-j},$$

where $h_{-j}$ does not depend on $\hat{c}_j$.

Differentiating with respect to $\hat{c}_j$ gives

$$\frac{du_j}{d\hat{c}_j} = (\hat{c}_j - c_j) \underbrace{\frac{\partial f_j(\boldsymbol{W}^{\star}(\hat{\mathbf{c}}))}{\partial \hat{c}_j}}_{\le 0} + \underbrace{\sum_{k \in \mathcal{S} \setminus \{j\}} \hat{c}_k \frac{\partial f_k(\boldsymbol{W}^{\star}(\hat{\mathbf{c}}))}{\partial \hat{c}_j}}_{\ne 0}$$

Since $\sum_{k \in \mathcal{S} \setminus \{j\}} \hat{c}_k \frac{\partial f_k}{\partial \hat{c}_j} \ne 0$, as the allocation to other buyers will change with $\hat{c}_j$. In order to have optimality, we would have $\hat{c}_j \ne c_j$.

$\square$

### C.3 SECTION 5 PROOFS

#### C.3.1 PROOF FOR MYERSON (SECTION 5.1) AND VCG PAYMENTS (SECTION 5.2

**Myerson Payments.** $f_j$ is monotonically non-increasing in $\hat{c}_j$, which we prove in Claim 1. We prove by illustration in Figure 7. When seller $j$ reports truthfully, the utility is equivalent to the size of the blue area in the leftmost figure ($s_1$). When seller $j$ under-reports, the utility becomes $s_1 - s_2$; when seller $j$ over-reports, the utility becomes $s_1 - s_3$. Thus, Myerson payment results in the highest utility when the report is truthful. By design, it is IR, as the utility $s_1$ is greater than 0.

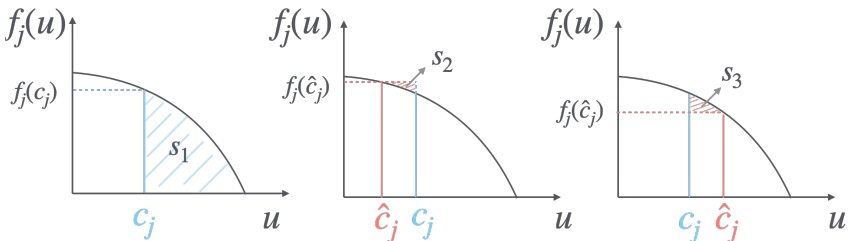

Figure 7: Myerson payment illustration.

**VCG Payments.** Let data seller $j$ have true cost $c_j$ with reported cost $t$. The other sellers have their reported costs $\hat{\mathbf{c}}_{-j}$.

We have seller $j$'s utility being

$$
\begin{aligned}
u_j(t) &= P_i^{vcg} - c_j f_j(\boldsymbol{W}^\star(t, \hat{\mathbf{c}}_{-j})) \\
&= \sum_{i \in \mathcal{B}} v_i(\boldsymbol{W}^\star(t, \hat{\mathbf{c}}_{-j})) - \sum_{k \in \mathcal{S}\backslash\{j\}} \hat{c}_k f_k(\boldsymbol{W}^\star(t, \hat{\mathbf{c}}_{-j})) - c_j f_j(\boldsymbol{W}^\star(t, \hat{\mathbf{c}}_{-j})) - h_{-j}
\end{aligned} \tag{18}
$$

where $h_{-j}$ is a term that is not dependent on $t$. Ignoring $h_{-j}$, this becomes the social welfare when the costs are $(c_j, \hat{\mathbf{c}}_{-j})$,

By definition, $\boldsymbol{W}^\star(t, \hat{\mathbf{c}}_{-j}))$ optimizes the social welfare when the costs are $(t, \hat{\mathbf{c}}_{-j})$. But seller $j$'s true utility depends on the actual cost $c_j$. Therefore, the only way seller $j$ ensures the system selects an allocation that maximizes their utility is to report truthfully, i.e., set $t = c_j$

### C.4 PROOF FOR CLAIM 4

*Proof.* For $\mathcal{B} \cup \mathcal{S}\backslash\{j\}$, we have $\boldsymbol{W}^\star(\infty, \boldsymbol{c}_{-j})$ being the maximum of $SW(\boldsymbol{c}_{-j})$. This gives

$$
\begin{aligned}
&\sum_{i \in \mathcal{B}} v_i(\boldsymbol{W}^\star(\infty, \boldsymbol{c}_{-j})) - \sum_{k \in \mathcal{S}\backslash\{j\}} c_k f_k(\boldsymbol{W}^\star(\infty, \boldsymbol{c}_{-j})) \geq \\
&\sum_{i \in \mathcal{B}} v_i(\boldsymbol{W}^{-j}(c_j, \boldsymbol{c}_{-j})) - \sum_{k \in \mathcal{S}\backslash\{j\}} c_k f_k(\boldsymbol{W}^{-j}(c_j, \boldsymbol{c}_{-j}))
\end{aligned} \tag{19}
$$

Plugging (19) in (8) gives the claimed inequality. □

#### C.4.1 PROOF FOR THEOREM 5.1

*Proof.* For the rest, we denote $f(\boldsymbol{W}^\star(u, \hat{\boldsymbol{c}}_{-j}))$ as $f_j(u)$.

From the IC condition, we have

$$
p(c_j) - c_j f_j(c_j) \geq p(c_j') - c_j f_j(c_j'), \tag{20}
$$

$$
p(c_j') - c_j' f_j(c_j') \geq p(c_j) - c_j' f_j(c_j), \tag{21}
$$

$$
c_j'[f(c_j) - f(c_j')] \geq p(c_j) - p(c_j') \geq c_j[f(c_j) - f(c_j')] \tag{22}
$$

Let $c'_j = u$, and $c_j = u + h$, with $h \to 0_+$, we have

$$uf'(u) \geq p'(u) \geq (u+h)f'(u) \tag{23}$$

and thus

$$uf'(u) = p'(u) \tag{24}$$

We have

$$p(u) = \int_\infty^u p'(u) = \int_\infty^u uf'(u)$$

$$= uf(u) - \int_\infty^u f(u)du \tag{25}$$

$$= uf(u) + \int_u^\infty f(u)du + C$$

Constrained to IR, we need to have $\int_u^\infty f(u)du + C \geq 0$. The integral is non-negative, so IR is satisfied iff $C \geq 0$. When $C = 0$, we have $p(u)$ equivalent to Myerson payment. $\square$

### C.4.2 DETAILED PROOF FOR THEOREM 5.2

*Proof.* Keep other sellers' reported $c_k$ ($k \neq j$) fixed, we have social welfare as a function of seller $j$' reported $c_j$

$$SW(\boldsymbol{W}^\star(\hat{c}_j, \hat{\boldsymbol{c}}_{-j})) = \sum_{i \in \mathcal{B}} v_i(\boldsymbol{W}^\star_{i,:}(\hat{c}_j, \hat{\boldsymbol{c}}_{-j})) - \sum_{j \in \mathcal{S}} \hat{c}_j f(\boldsymbol{W}^\star_{:,j}(\hat{c}_j, \hat{\boldsymbol{c}}_{-j})) \tag{26}$$

As $\boldsymbol{W}^\star$ is chosen by the system to maximize the reported social welfare, we have

$$\frac{\partial v_i(\boldsymbol{W}^\star_{i,:})}{\partial w^\star_{i,j}} - \hat{c}_j \frac{\partial f(\boldsymbol{W}^\star_{:,j})}{\partial w^\star_{i,j}} = 0, \qquad \forall w_{i,j} \tag{27}$$

We further show that $\frac{\partial SW(\boldsymbol{W}^\star(\hat{c}_j, \hat{\boldsymbol{c}}_{-j}))}{\partial \hat{c}_j} = -f_j$

$$\frac{\partial SW(\boldsymbol{W}^\star(\hat{c}_j, \hat{\boldsymbol{c}}_{-j}))}{\partial \hat{c}_j} = \sum_{i \in \mathcal{B}} \sum_{k \in \mathcal{S}} \frac{\partial v_i(\boldsymbol{W}^\star_{i,:})}{\partial w^\star_{i,k}} \frac{\partial w^\star_{i,k}}{\partial \hat{c}_j}$$

$$- f(\boldsymbol{W}^\star_{:,j}) - \sum_{k \in \mathcal{S}} \hat{c}_k \sum_{i \in \mathcal{B}} \frac{\partial f_k(\boldsymbol{W}^\star)}{\partial w^\star_{i,k}} \frac{\partial w^\star_{i,k}}{\partial \hat{c}_j} \tag{28}$$

$$\overset{(27)}{=} -f(\boldsymbol{W}^\star_{:,j})$$

Thus, we can calculate the integral part of the Myerson payment as

$$\int_{c_j}^\infty f(\boldsymbol{W}^\star_{:,j}(u, \hat{\boldsymbol{c}}_{-j}))du = SW(\boldsymbol{W}^\star(\hat{c}_j, \hat{\boldsymbol{c}}_{-j})) - SW(\boldsymbol{W}^\star(\infty, \hat{\boldsymbol{c}}_{-j})) \tag{29}$$

As Myerson payment will incentivizes truthful reporting, we have

$$P_{\to j}^{MS} = c_j f(\boldsymbol{W}^\star_{:,j}(c_j, \boldsymbol{c}_{-j})) + \int_{c_j}^\infty f(\boldsymbol{W}^\star_{:,j}(u, \boldsymbol{c}_{-j}))du$$

$$= c_j f(\boldsymbol{W}^\star_{:,j}(c_j, \boldsymbol{c}_{-j})) + \sum_{i \in \mathcal{B}} v_i(\boldsymbol{W}^\star_{i,:}(c_j, \boldsymbol{c}_{-j})) - \sum_{j \in \mathcal{S}} c_j f(\boldsymbol{W}^\star_{:,j}(c_j, \boldsymbol{c}_{-j}))$$

$$- \left[ \sum_{i \in \mathcal{B}} v_i(\boldsymbol{W}^\star_{i,:}(\infty, \boldsymbol{c}_{-j})) - \sum_{j \in \mathcal{S}} c_j f(\boldsymbol{W}^\star_{:,j}(\infty, \boldsymbol{c}_{-j})) \right] = P_{\to j}^{VCG} \tag{30}$$

$\square$

### C.4.3 PROOF OF THEOREM 5.3

*Proof.* As $P_{\to j}^{MC}$ is the minimal payment rule that fulfills IC and IR for sellers, we must have $P_{\to j}^{MC} < P_{\to j}^{vcg}$. Claim 4 indicates

$$P_{\to j}^{MS} \le P_{\to j}^{vcg} \le \sum_{i \in \mathcal{B}} v_i(\boldsymbol{W}^\star(c_j, \boldsymbol{c}_{-j})) - v_i(\boldsymbol{W}^{\star-j}(c_j, \boldsymbol{c}_{-j})) \tag{31}$$

Following the choice of $\eta_{ij}$, we have

$$P_{i \to j} \le v_i(\boldsymbol{W}^\star(c_j, \boldsymbol{c}_{-j})) - v_i(\boldsymbol{W}^{\star-j}(c_j, \boldsymbol{c}_{-j})) \tag{32}$$

We can thus bound the payment from buyer $i$ by

$$P_{i \to} = \sum_j \eta_{ij} P_{\to j} \le \sum_j v_i(\boldsymbol{W}^\star(\boldsymbol{c})) - v_i(\boldsymbol{W}^{\star-j}(\boldsymbol{c})) \overset{\text{subadditive}}{\le} v_i(\boldsymbol{W}^\star(\boldsymbol{c})) - v_i(\boldsymbol{0}) \tag{33}$$

The final inequality is a consequence of the sub-additivity assumption, under which the aggregate contribution of data sellers is bounded above by the sum of their separate contributions. □

## D ADDITIONAL EXPERIMENTAL RESULTS

### D.1 MEAN ESTIMATION MARKETS

#### D.1.1 EXPERIMENTAL SETUP

Choose the true data dimension as 3, that is $\boldsymbol{\mu} \in \mathbb{R}^3$. We first randomly sample true means $\{\mu_i\}_{i \in |\mathcal{B}|}$ and $\{\mu_j\}_{j \in |\mathcal{S}|}$, where $|\mathcal{B}| = 5$ and $|\mathcal{S}| = 10$. The true means are sampled from a normal distribution with mean 1 and variance 1. By assuming that sellers have unbiased local mean estimates, we only need to sample local variations $\{\sigma_j^2\}_{j=1}^{|\mathcal{B}|}$, which is sampled from $\text{Unif}[0, 1]$. The true unit costs $\boldsymbol{c}$ are sampled uniformly from 1 to 10.

#### D.1.2 COMPARISON OF DIFFERENT TRUTHFUL PAYMENT RULES

We focus on the seller with index $j = 0$ and analyze its payment and cost while keeping the reported costs of other sellers fixed. This allows us to easily compute and compare different payment rules as we vary the true $c_j$. In the mean estimation market, $\boldsymbol{W}^\star(\hat{c}_j, \hat{\boldsymbol{c}}_{-j})$ can have negative entries. Thus, $\sum_{k \neq j} f_k(\boldsymbol{W}^\star(\hat{c}_j, \hat{\boldsymbol{c}}_{-j}))$ can decrease or increase with $c_j$, while $f_j(\boldsymbol{W}^\star(\hat{c}_j, \hat{\boldsymbol{c}}_{-j}))$ always decrease with $c_j$. This is shown in Figure 8, where the results of two randomly generated mean estimation markets are presented. Since $\boldsymbol{W}$ lies in an unconstrained domain, we have VCG payment equivalent to Myerson payment, as shown in Figure 9a, where the value mismatch is due to numerical issues.

#### D.1.3 MIS-REPORTING FOR UNTRUTHFUL PAYMENT RULES

Let true $c_j = 5$. We compute the payment values and resulted utility when varying the reported $\hat{c}_j$. Given our choice of $\boldsymbol{W}$, social cost is always minimized at $\hat{c}_j = c_j$. However, with LOO and Shapley payment rule, the resulted utilities peak at a different value than $c_j$, as shown in Figure 4.

**LOO Payment Rule.** To maximize its utility, seller $j$ will mis-report $\hat{c}_j = 9$. The resulting $PoA = SW(\boldsymbol{W}^\star(6, \boldsymbol{c}_{-j}))/SW(\boldsymbol{W}^\star(5, \boldsymbol{c}_{-j})) = 1.02$.

**Shapley Payment Rule.** To maximize its utility, seller $j$ will mis-report $\hat{c}_j = 8$. The resulting $PoA = SW(\boldsymbol{W}^\star(8, \boldsymbol{c}_{-j}))/SW(\boldsymbol{W}^\star(5, \boldsymbol{c}_{-j})) = 1.08$

### D.2 DATA MIXTURE MARKET

Our framework extends to LLM training as well. For simplicity, we consider a single buyer that is seeking for purchasing data from different data sellers to arrive at a good pretrained model. From Ye et al. (2024), the relationship between proportions of data mixtures and the validation loss in the

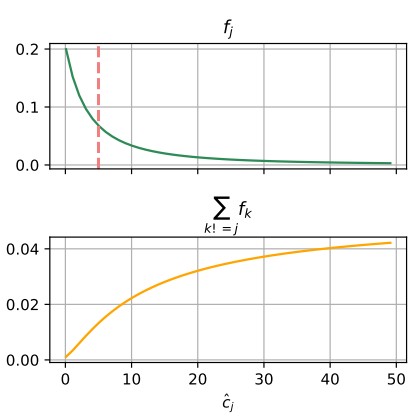
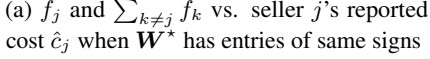
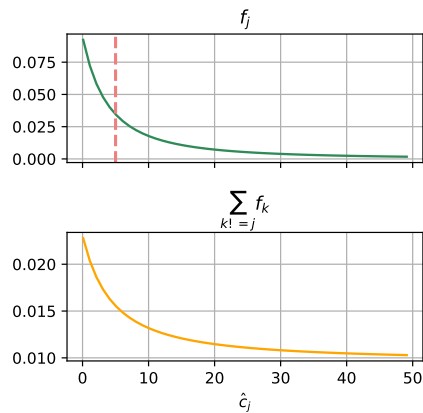

(a) $f_j$ and $\sum_{k \neq j} f_k$ vs. seller $j$'s reported cost $\hat{c}_j$ when $\boldsymbol{W}^\star$ has entries of same signs

(b) $f_j$ and $\sum_{k \neq j} f_k$ vs. seller $j$'s reported cost $\hat{c}_j$ when $\boldsymbol{W}^\star$ has entries of different signs

Figure 8: How data sharing cost changes with seller $j$'s reported cost factor $\hat{c}_j$ for Mean Estimation Market.

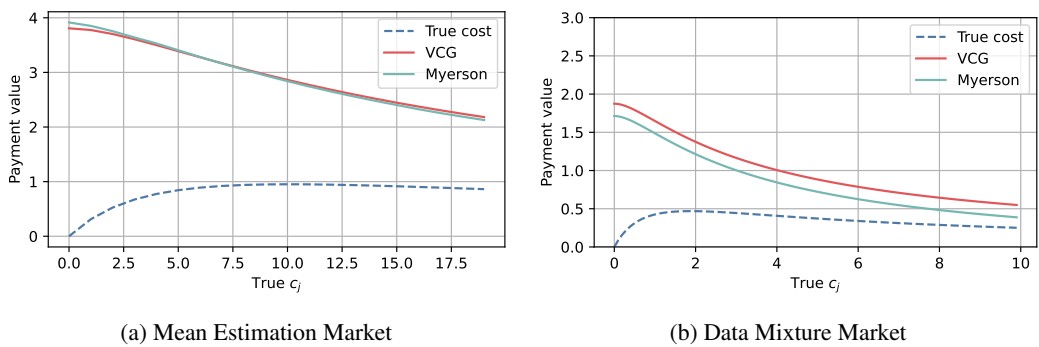

(a) Mean Estimation Market

(b) Data Mixture Market

Figure 9: Comparison of different truthful payment rules with respect to varying true cost $c_j$s

target domain can be modeled via (34), assuming fixed token budgets. $b$ and $k$ are constant scalars, and $\boldsymbol{t} = [t_i] \in \boldsymbol{R}^{|\mathcal{S}|}$ captures the how the training data of seller $j$ helps reduce validation loss on the target domain. $\boldsymbol{w} \in \boldsymbol{R}^{|\mathcal{S}|}$ denotes the proportion of pertaining data from each seller $j$.

$$L = b + k \exp \left( \sum_{j \in \mathcal{S}} t_j w_j \right) \tag{34}$$

Following our framework, the data mixtures can be determined from minimizing the social cost objective in (35).

$$\boldsymbol{w} \in \arg\min_{\boldsymbol{w}} L(\boldsymbol{w}) + \sum_{j \in \mathcal{S}} \hat{c}_j w_j^2 \tag{35}$$

In practice, (34) can be estimated using a small LM on small datasets. To pre-determine the data mixtures via a smaller LM is a standard practice in LLM pre-training. Here we randomly sample some values to be $(k, b, \boldsymbol{t})$. Given that $\boldsymbol{W}$ here are row-stochastic, the rank of truthful payment methods again confirms our theoretical analysis, as shown in Figure 9b.

## D.3 RAG MARKET

### D.3.1 EXPERIMENTAL DETAILS

**Dataset:** MedQuAD dataset Ben Abacha & Demner-Fushman (2019), which is a medical dataset containing 47,457 QA pairs.

**Procedures:**

- Create a vector store using answers from MedQuAD dataset using Faiss library (Douze et al., 2024). The embedding model used is `text-embedding-3-large` from OpenAIEmbeddings.

- Generate user queries from the questions from MedQuAD, the goal is to make the similarity search less straightforward, so that it requires some semantic understanding to retrieve the correct document.

- Conduct Top-$n$ similarity-based retrieval and cost-aware reranking. Get the final $k$ documents.

- Feed the retrieved $k$ documents into the LLM generator, which is `Qwen2.5-3B`, Yang et al. (2024), and generate the context-enhanced answer.

- The LLM judge (`DeepSeek-R1`, DeepSeek-AI (2025)) evaluates the answer given the query and return a score between 0 and 10.

- Calculate the different payments for the documents in $\mathcal{C}_n$.

In practice, due to the combinatorial complexity of the decision space—specifically, the $\binom{n}{k}$ possible combinations from $\mathcal{C}_n$ —the reranking process is simplified not to consider interactions among the selected $k$ documents. Each of the $n$ documents is scored independently, and the top $k$ documents favored by the LLM judge are selected. Our experiment follows the same setup.

### D.3.2 PROMPT FOR THE RAG MARKET

Listing 1: LLM as a Judge Prompt

```
You will be given a user_question and system_answer couple.
Your task is to provide a 'total rating' scoring how well the
    system_answer answers the user concerns expressed in the
    user_question.
Give your answer as a float on a scale of 0 to 10, where 0 means
    that the system_answer is not helpful at all, and 10 means
    that the answer completely and helpfully addresses the
    question.

Provide your feedback as follows:

Feedback:::
Total rating: (your rating, as a float between 0 and 10)

Now here are the question and answer.

Question: {question}
Answer: {answer}

Feedback:::
Total rating:
```

Listing 2: Question Generation Prompt

```
You emulate a user of our medical question answering application.
Formulate 4 questions this user might ask based on a provided
    disease.
```

```
Make the questions specific to this disease.
The record should contain the answer to the questions, and the
    questions should
be complete and not too short. Use as few words as possible from
    the record.

The record:

question: {question}
answer: {answer}
source: {source}
focus_area: {focus_area}

Provide the output in parsable JSON without using code blocks:

{{"questions": ["question1", "question2", ..., "question4"]}}
```

### D.3.3 ARE LLM JUDGES TRUSTWORTHY?

We randomly sampled 200 queries from the MedQuAD dataset, each associated with a known ground truth answer. To prevent trivial matches based on string similarity, we rewrote the queries so that the retrieval step could not rely solely on lexical overlap. The retrieval corpus consisted of all answer entries from MedQuAD. For this experiment, we used top-1 retrieval only, and the retrieved document was then used for answer generation. Among the 200 retrievals, 158 correctly matched the ground truth documents.

Note that, besides the ground-truth document, other documents in the corpus may still provide partial answers to the query. However, we expect the LLM Judge to assign higher scores when the correct (ground-truth) context is retrieved—and this is exactly what we observe: the average score was 8.23 when responses used ground-truth context, compared to 7.04 when they did not. This suggests that LLM Judge scores meaningfully reflect the quality of the retrieved documents.

### D.4 PAYMENT CALCULATION PSEUDOCODE IN RAG MARKET

$P_{\to j}^{\text{Shapley}}$ is not included in the pseudo code, due to its complex format. To get $P_{\to j}^{\text{Shapley}}$, one can first calculate $P_{\to j}^{\text{LOO}}$ for all subsets of $[n]\backslash\{j\}$, and calculate the average.

---

**Algorithm 1** Payment calculation in a RAG Market with $k = 1$

---

**Require:** LLM judge scores $\mathbf{s} = [s_i]_{i=1}^n$, Costs $\mathbf{p} = [p_i]_{i=1}^n$ with $n \geq 2$
1: {Compute Social welfare $\phi$}
2: **for** $i \leftarrow 1$ to $n$ **do**
3:     $\phi_i = s_i - p_i$
4: **end for**
5: {Pick winner and record its price}
6: $j \leftarrow \arg\max_i \phi_i$
7: $c_j \leftarrow p_j$
8: {Sort $\phi$ in descending order}
9: (Sorted SW $\widetilde{\phi}$, SortedIdx $I$) $\leftarrow$ sorted($\phi$, descending)
10: {Myerson payment}
11: $P_{\to j}^{\text{Myerson}} \leftarrow c_j + (\widetilde{\phi}_1 - \widetilde{\phi}_2 - c_j) = \widetilde{\phi}_1 - \widetilde{\phi}_2$
12: $P_{\to j}^{\text{VCG}} \leftarrow \widetilde{\phi}_1 - \widetilde{\phi}_2$
13: $P_{\to j}^{\text{LOO}} \leftarrow s[I[1]] - s[I[2]]$

---

### D.5 ILLUSTRATIVE EXAMPLES FOR THE RAG MARKET

**Example 1** (Truthful LOO Payment with Low Unit Costs). *For document 1, it will report truthfully, as $P_1^{LOO} = v(doc\ 1 \cup doc2) - v(doc2) = 0.2$ is irrelevant to document 1 owner's reported cost $\hat{c}_1$ and greater than document 1's unit cost. The owner of Document 2 also has no incentive to misreport, since even reporting the minimum possible cost $\hat{c}_2 = 0$ does not lead to retrieval; its utility remains lower than that of Document 1 (0.8).*

|      | LLM judge score $v$ | unit cost $c$ | SW $\phi$ |
|------|------|------|------|
| *doc1* | *0.9* | *0.1* | *0.8* |
| *doc2* | *0.7* | *0.05* | *0.65* |

**Example 2** (Untruthful LOO Payment with High Unit Costs). *Now imagine the unit data costs are higher. In this scenario, document 1 should still get retrieved if both report truthfully. Yet, $P_1^{LOO} = v(doc\ 1 \cup doc2) - v(doc2) = 0.2 < c_1$. That is, the payment is not IR for document 1 owner. Document 1 owner will over-report the cost, so document 1 will never get retrieved. Since the payment is negative, the owner of document 2 also has no incentive to participate and will prefer to over-report their cost. As a result, neither document will be retrieved due to over-reporting.*

|      | LLM judge score $v$ | unit cost $c$ | SW $\phi$ |
|------|------|------|------|
| *doc1* | *0.9* | *0.3* | *0.6* |
| *doc2* | *0.7* | *0.2* | *0.5* |

**Example 3** (Top2 Retrieval). *If all document owners report truthfully, both doc1 and doc2 will be retrieved. $P_1^{LOO} = v(doc\ 1 \cup doc2) - v(doc2 \cup doc3) = 0.2 > c_1$ and $P_2^{LOO} = v(doc\ 1 \cup doc2) - v(doc1 \cup doc3) = 0.1 > c_2$. In this scenario, no document owner will lie about their true costs. Shapley payment is the same as the LOO payment in this case.*

*For VCG payment, following the (8), we have*

$$P_1^{VCG} = u(doc\ 1 \cup doc2) - u(doc\ 2 \cup doc3) + c_1 = 0.3 \tag{36}$$

$$P_2^{VCG} = u(doc\ 1 \cup doc2) - u(doc\ 1 \cup doc3) + c_2 = 0.2 \tag{37}$$

*For Myerson payment, we first need to decide the exact $\hat{c}'_j$ for each document owner above which it will no longer get retrieved. For doc1, it is $0.3$, and for doc1 it is $0.2$. So we have*

$$P_1^{LOO} = c_1 + (\hat{c}'_1 - c_1) = 0.3, \quad P_2^{LOO} = c_2 + (\hat{c}'_2 - c_2) = 0.2 \tag{38}$$

*We see that VCG and LOO payments are still equivalent.*

Table 1: Costs for each doc

|      | unit cost $c$ |
|------|------|
| *doc 1* | *0.1* |
| *doc 2* | *0.05* |
| *doc 3* | *0.1* |

Table 2: Utility for each 2-doc retrieval

|      | LLM judge score $v$ | true costs | SW $\phi$ |
|------|------|------|------|
| *doc1, doc2* | *0.9* | *0.15* | *0.75* |
| *doc2, doc3* | *0.7* | *0.15* | *0.55* |
| *doc1, doc3* | *0.8* | *0.2* | *0.6* |

## E CHALLENGES WITH PRIVATE BUYER PERFORMANCE

So far, we have looked into the scenario where buyers' performance function is assumed to be known. In practice, such functions are usually private, especially when entering into a new market with no historical records. Or when sharing the performance functions violates the buyer's privacy. In such cases, we show that it is impossible to design any payment rule that simultaneously achieves IC, IR, WBB, and SE. In fact, the social cost may be arbitrarily far from the efficient solution. This result is similar in spirit to the celebrated Myerson–Satterthwaite theorem (Myerson & Satterthwaite, 1983) proving the impossibility of efficient bilateral trade (double auctions). The negative results open up a new challenge for designing valuation and pricing rules for data markets.

### E.1 An Impossibility Result

**Theorem E.1** (cr. Holmström (1979)). *Assume all participants' valuation $v_i \in \mathcal{V}_i$. If $\mathcal{V} = \times \mathcal{V}_i$ is a convex domain, then Groves mechanism that is defined by the allocation rule*

$$\boldsymbol{W}^\star \in \arg\max_{\boldsymbol{W} \in \mathcal{W}} \sum_i \hat{v}_i(\boldsymbol{W})$$

*and payment rule*

$$p_i = h_i(-\hat{v}_i) - \sum_{j \neq i} \hat{v}_j(\boldsymbol{W})$$

*is the unique IC mechanism, up to the choice of $h_i$*

**Theorem E.2** (Impossibility Result). *When buyers and sellers both have their private types to report, no single mechanism can simultaneously satisfy IR, WBB, IC and SE. Further, any mechanism that ensures WBB, IR, and IC can result in arbitrarily poor social efficiency.*

*Proof.* We consider a simple one-buyer-one-seller case, where the reported valuations are

$$\hat{v}_b = \hat{l}_b(0) - \hat{l}_b(w_{bs}^\star), \ \hat{v}_s = -\hat{c}_s f(w_{bs}^\star) \tag{39}$$

where $w_{bs}^\star \in \arg\min \hat{l}_b(0) - \hat{l}_b(w_{bs}^\star) - \hat{c}(w_{bs}^\star)$. Truthfulness requires Groves payment rule, thus $p_b = h(v_s) - v_s$ and $p_s = h(v_b) - v_b$. With truthful reporting, we have social efficiency directly follows as the coordinator chooses socially optimal $\boldsymbol{W}^\star$.

Now we check IR and WBB conditions, with IR, we have

$$v_b + v_s \geq \max(h(v_b), h(v_s)) \tag{40}$$

With WBB, we have

$$v_b + v_s \leq h(v_b) + h(v_s) \tag{41}$$

Let $v_b \to 0$, we have $v_s \geq \max(h(0), h(v_s))$. This implies $v_s \geq h(v_s)$. Analogously, $v_b \geq h(v_b)$.

Let $\delta_b = v_b - h(v_b)$ and $\delta_s = v_s - h(v_s)$, we have $\delta_b \geq 0$ and $\delta_s \geq 0$. Plugging in (41), we have

$$\delta_b + \delta_s + h(v_b) + h(v_s) \leq h(v_b) + h(v_s)$$

which suggests $\delta_b + \delta_s \leq 0$. It can only be $\delta_b = \delta_s = 0$. That is, $p_b = p_s = 0$. Both the buyer and the seller only take their own valuations into account. As the seller's valuation will be negative as long as $w_{bs}^\star > 0$, constrained to IR conditions, there will not be trade happening.

Now let's check the Price of Anarchy (PoA):

$$PoA = \frac{l_b(0)}{\min_{w_{bs}} l_b(w_{bs}) + c_s f(w_{bs})} \tag{42}$$

PoA can be arbitrarily large if $\min_{w_{bs}} l_b(w_{bs}) + c_s f(w_{bs}) \to 0$, when happens when seller $s$ has exactly the data buyer $b$ needs, which makes buyer $b$'s loss goes to 0, and the cost $c_s$ is 0.

$\square$

