# OpenReview forum: "Do Data Valuations Make Good Data Prices?"
_ICLR.cc/2026/Conference — Submitted to ICLR 2026_

### Official Review · Reviewer_3FgT · 2025-10-31

**Soundness:** 3
**Presentation:** 2
**Contribution:** 2
**Rating:** 4
**Confidence:** 3

**Summary:**

The paper considers the setting where data owners have private heterogeneous costs and seek to devise a mechanism that maximizes the social welfare while incentivising data owners to truthfully report their data unit costs. The paper shows that LOO and Data Shapley do not incentivise truthful reporting. The paper then adapt the Myerson and VCG payment rules and show that they are incentive compatible and can ensure individual rationality under some conditions. The paper demonstrates one experiment/application with a market for retrieval augmented generation.

**Strengths:**

1. The problem considered of incentivising data owners to truthfully report their data unit costs and maximise social/individual welfare is important and have real life applications.
2. The paper is generally easy to follow. The figures are helpful.

**Weaknesses:**

1. It is unclear what are the technical challenges involved and how this work novelly addresses them. It will be helpful to clarify them.
    * Myerson lemma and VCG auction are known to be incentive compatible. These are well-known results from game theory.
    * There are other related works that also ensures IC. What sets this work apart? The data markets paragraph in Sec 2 should be more in depth.
    * Data Shapley and LOO are designed to ensure fairness and be cost-agnostic. Naturally, they do not incentivise truthful reporting of cost. It should also be clarified (and justified) that the Data Shapley and LOO in this paper differs from the original as social welfare is considered.
2. There should be further justification of the modeling framework in Sec 3, e.g., specific use cases where the buyer/seller interactions can be modelled by $W$.
3. The experiment considered is simple and each $w$ is either $0$ or $1$.


[1] Cong, M., Yu, H., Weng, X., Qu, J., Liu, Y., & Yiu, S. M. (2020). A VCG-based fair incentive mechanism for federated learning. arXiv preprint arXiv:2008.06680.
[2] Tang, X., Yu, H., Li, X., & Kraus, S. (2024). Intelligent agents for auction-based federated learning: A survey. arXiv preprint arXiv:2404.13244.

**Questions:**

1. What are some specific use cases where the buyer/seller interactions can be modelled by $W$?
2. What is the challenge in this work when adapting Myerson lemma and VCG auction for data valuation?
3. Elaborate on “We note that most of these works often assume known simple structured valuation functions and combinatorial allocations, not suitable for machine learning worlds where the information sharing can happen in continuous space and the buyers’ valuations are directly connected to model performances, which is our focus.” With specific examples and citations.

---

> ### Author Response · Authors · 2025-11-21
>
> Thanks for your insightful feedback. We address your raised concerns as follows:
>
> **W1**: 1) The goal of our paper is to highlight the two different concepts: data pricing and data valuation. Data valuation methods are often used for data pricing, and we want to point out that they are not suitable if the data sellers are strategic.
> 2) And yes, there are many works that ensure IC, but they function in a hyper-synthetic setup. The innovation/contribution of our work lies in a) providing a realistic data market modeling framework;  b) pointing out a different consideration for data-centric ML.
>
> **W2 & Q1**: We list use cases where both the buyers’ performance function and sellers’ data sharing costs can be modeled by W.
>
> - Acquiring Data from Multiple Domains: When conducting pre-training runs for LLMs,
> it is usually needed to decide on the data mixtures from different domains/sellers. With
> a larger $w_{i,j}$, buyer i benefits from more data from seller j, and for seller j, the costs for
> collecting and preparing the data also increase. $\mathbf{W} \in \{\mathbf{W}: w_{i,j} \geq 0 \ and \ \sum_{j \in S} w_{i,j} = 1\}$.
>
> -  Differential Private Model Sharing: When sharing models, seller j shares a noisy version
> of local model ($\theta_j +\epsilon_{i,j}$) with buyer i, where $\epsilon_{i,j} \sim \mathcal{N}(0,1)$. As $|w_{i,j}|^2$  increases, less
> noise is added to the model shared by seller j, resulting in a higher degree of privacy. $\mathbf{W} \in \mathbb{R}^{N \times N}$.
>
> -  Data Compensation in RAG: where $w_{ij}$ denotes if document $j$ is retrieved for query $i$.
>
> **W3**: We have three experiments studied in the paper: 1) RAG market, where $w_{i,j}$ is 0 or 1 by nature; 2) Mean estimation market, where $w_{i,j} \in \mathbb{R}$; 3) Data Mixture Market, where  $W$ is row stochastic (Appendix D.2). We highlight the RAG scenario because it closely reflects a real-world application.
>
> **Q2**: We mostly adopted the default Myerson and VCG formulas. The challenge lies more in 1) coming up with a realistic data market modeling framework, 2) finding the exact relationship between $\hat{c}\_j$ and $P\_{\rightarrow j}$, as this requires optimizing reported social welfare. Most of the time, a closed-form expression is not feasible.
>
> **Q3**:  We have updated our script with the specified citations.
>
> If you have any further questions or ideas, please let us know. We’d be glad to share additional supporting information.

---

### Official Review · Reviewer_C5p4 · 2025-10-31

**Soundness:** 3
**Presentation:** 3
**Contribution:** 2
**Rating:** 4
**Confidence:** 3

**Summary:**

This paper addresses a challenge in data markets for large language models (LLMs): designing truthful and efficient payment mechanisms to compensate data contributors, as traditional data valuation methods fail to account for market dynamics. It demonstrates that popular data valuation methods—Leave-One-Out (LOO) and Data Shapley—incentivize strategic misreporting (over-reporting/under-reporting) when used as pricing rules, reducing social welfare. To solve this, the paper adapts two classic mechanism design frameworks—Myerson payment rule and Vickrey-Clarke-Groves (VCG) mechanism—to data markets, proving they satisfy incentive compatibility (IC), individual rationality (IR), and social efficiency (SE). Additionally, it identifies that when buyers’ utility functions are subadditive, payments can be distributed to ensure buyers’ IR, and in unconstrained allocation scenarios, Myerson and VCG payments are equivalent.

**Strengths:**

1-It targets a timely, high-stakes problem—LLM data sourcing and contributor compensation—amid rising copyright litigation and data scarcity concerns. Unlike prior work focusing on data valuation for ML interpretability, it centers on market design, filling a critical gap between theoretical mechanism design and real-world data trading.

2-The paper’s framework is mathematically sound: it formalizes allocation, utility, and social welfare, provides complete proofs for key claims (e.g., Myerson’s minimality, VCG’s upper bound) in appendices, and derives closed-form solutions for mean estimation markets, ensuring theoretical trustworthiness.

3-The impossibility result clarifies what is unachievable (e.g., private buyer valuations), avoiding overpromises and guiding future research.

**Weaknesses:**

1-Unrealistic Assumption of Known Buyer Valuations: The main analysis assumes buyers’ performance functions (\(v_i\)) are publicly known, which contradicts real-world practice—LLM entities rarely disclose model performance gains from external data (to protect competitive advantage). While the paper acknowledges this and proves an impossibility result for private buyer valuations, it offers no mitigation strategies (e.g., approximate mechanisms, partial valuation disclosure), limiting the framework’s real-world applicability.

2-The paper only guarantees buyers’ IR when \(v_i\) is subadditive (diminishing returns from data; Theorem 5.3) but provides no solution for superadditive scenarios (e.g., complementary medical data, where combining datasets drives large performance gains). Superadditivity is common in LLM data sourcing, yet the paper labels this an "open question" (Remark 3) without exploratory analysis, weakening the framework’s comprehensiveness.

3-While Myerson is theoretically optimal, its integral calculation (\(\int_{\hat{c}_j}^{\infty} f_j(W^*(u,\hat{c}_{-j})) du\)) depends on optimizing social welfare for all \(u > \hat{c}_j\). For complex \(v_i\) (e.g., non-linear LLM loss functions) or large seller sets, this becomes computationally prohibitive. The paper mentions this issue but offers no approximations or efficient implementations, limiting Myerson’s practical use for large-scale data markets.

4-Most experiments focus on single-buyer settings (e.g., RAG with one user query) or small buyer sets (mean estimation with |B|=5). Multi-buyer data markets (e.g., multiple LLMs competing for the same medical dataset) introduce externalities (e.g., price competition) that the framework claims to handle but does not validate experimentally, raising doubts about scalability.

**Questions:**

1-Unrealistic Assumption of Known Buyer Valuations: The main analysis assumes buyers’ performance functions (\(v_i\)) are publicly known, which contradicts real-world practice—LLM entities rarely disclose model performance gains from external data (to protect competitive advantage). While the paper acknowledges this and proves an impossibility result for private buyer valuations, it offers no mitigation strategies (e.g., approximate mechanisms, partial valuation disclosure), limiting the framework’s real-world applicability.

2-The paper only guarantees buyers’ IR when \(v_i\) is subadditive (diminishing returns from data; Theorem 5.3) but provides no solution for superadditive scenarios (e.g., complementary medical data, where combining datasets drives large performance gains). Superadditivity is common in LLM data sourcing, yet the paper labels this an "open question" (Remark 3) without exploratory analysis, weakening the framework’s comprehensiveness.

3-While Myerson is theoretically optimal, its integral calculation (\(\int_{\hat{c}_j}^{\infty} f_j(W^*(u,\hat{c}_{-j})) du\)) depends on optimizing social welfare for all \(u > \hat{c}_j\). For complex \(v_i\) (e.g., non-linear LLM loss functions) or large seller sets, this becomes computationally prohibitive. The paper mentions this issue but offers no approximations or efficient implementations, limiting Myerson’s practical use for large-scale data markets.

4-Most experiments focus on single-buyer settings (e.g., RAG with one user query) or small buyer sets (mean estimation with |B|=5). Multi-buyer data markets (e.g., multiple LLMs competing for the same medical dataset) introduce externalities (e.g., price competition) that the framework claims to handle but does not validate experimentally, raising doubts about scalability.

---

> ### Author Response · Authors · 2025-11-21
>
> We thank the reviewer for their thoughtful and insightful feedback.
>
> We want to emphasize again that this is not a methodology paper, and we are not proposing a new truthful payment method. The goal of this paper is to _revisit_ the well-established data valuation methods. While they are good for data selection, they are not suitable for data compensation. Even though both data selection and compensation are dependent on the importance of data, data compensation has another important consideration, which is the incentives of data owners to report truthfully.
>
> **W1**: We acknowledge that known buyer valuations are a rather unrealistic assumption. In some cases, however, such a valuation function can be estimated. Essentially, the valuation function depicts how a buyer’s value changes with different data sellers’ data. For example, when choosing the best data mixture in the pretraining time, one can always fit a small proxy model to get the data mixture scaling law [1] (see our Appendix D.2). In the RAG case, this can be modelled with LLM as a judge, as shown in our paper. We further checked that the LLM-as-a-judge result aligns with the ground truth values, see evidence in Appendix D.3.3.
>
> **W2**: For superadditive scenarios, it could be that budget balance can no longer be fulfilled, and external subsidies to buyers are necessary.
>
> **W3**: You are correct that computing the Myerson payment can be demanding when $W$ is continuous. When $W$ is discrete, as noted in Lines 471-476, the integral becomes a step function and is straightforward to evaluate. Even in the continuous setting, once the buyer's valuation is known, we can approximate the integral by sampling several $\tilde{c}_j$ values from the interval $[c_j, \infty)$; increasing the number of samples improves accuracy. This is precisely the approach we used for the mean-estimation example. We will clarify this point in the revised version.
>
> **W4**: In our RAG market setup, multi-buyer would mean multiple user queries, instead of multiple LLMs. We consider a single-buyer scenario, since in practice users are charged per query (i.e., per purchase), as already pointed out in Line 406-407. The price competition scenario could be interesting to study, but it is beyond the scope of this work.
>
> [1] Ye et al. Data Mixing Laws: Optimizing Data Mixtures by Predicting Language Modeling Performance

---

### Official Review · Reviewer_avQc · 2025-10-31

**Soundness:** 3
**Presentation:** 3
**Contribution:** 3
**Rating:** 6
**Confidence:** 2

**Summary:**

This paper tackles a timely problem of designing a fair data-trading market where data creators are compensated fairly and data buyers are able to improve model performance.

Traditional data attribution methods like Leave-One-Out and Data Shapley incentivise data owners to misreport their data prices. This paper proposed to use the well-known Myerson payments and VCG mechanism to design fair data trading framework. The core analysis shows: (i) the Myerson payment rule yields the minimum possible payment, making it optimal from the buyer’s standpoint; and (2) when allocations are made to maximize overall market welfare in an unconstrained setting, the VCG and Myerson payments coincide.

As an example, the authors demonstrate contributor compensation in an LLM based retrieval-augmented generation (RAG) marketplace for medical question answering.

**Strengths:**

1. The problem of fair and truthful compensation for data owners is extremely timely and impactful given the increasing infringement of copyright laws by large AI corporations.

2. While the strength of this work is not necessarily a contribution to the mechanism design literature, its novelty lies in adaptation of well-known VCG and Myerson mechanism to an important problem.

3. The application demonstrated in a RAG setting is practically useful.

**Weaknesses:**

1. For a non-expert reader, the theoretical parts of the paper are hard to read and understand especially since most proofs are deferred to the Appendix. While I have some background in mechanism design, I was not able to understand and verify all the proofs.

2. The paper assumes that buyers' valuations are known, i.e., how valuable is a certain data point to improving a give model. How practical is this assumption given that most closed-source companies barely reveal anything about their modelling process? Some justification for this assumption should help.

3. While the pretext of the paper is designing fair data-markets, there is a gap between the theoretical results shown and the practical application demonstrated. The RAG application is more like an inference setup while a lot of the discussion leading up to the theoretical results seems catered to selecting pre-training data for models. I am not sure if the proposed VCG mechanism generalises to a pre-training setup since it is nearly impossible to quantify impact of (absence of) one data point in the train set.

**Questions:**

1. Can you provide some justification to the assumption of known buyer valuations? Perhaps some practical examples will help.

2. Have you thought about how this kind of a setup can be applied to a pre-training setup? Perhaps in each epoch, the buyer can choose data that is most helpful for the model to improve and the data buyers are compensated accordingly? In earlier epochs, perhaps simpler kinds of data (like school level math) will be compensated more than in later epochs, more advanced kinds of data (like olympial level math) can be compensated more?

---

> ### Author Response · Authors · 2025-11-21
>
> Thanks for your insightful feedback. We address your raised concerns as follows:
>
> **W1**: Thanks for the comment. We had to defer a lot of proofs to the Appendix due to the space constraints; however, we are quite confident that they should be correct.
>
> **W2**: Please see the first point of our global response.
>
> **W3**: Thanks for pointing out the potential application of our market modeling in pretraining. For the pretraining, we cannot quantify the impact of (absence of) one data point in the train set; however, we can only quantify the impact of an entire domain, as shown in our Appendix D.2 Data Mixture Market. Please note that the goal of this work is not to measure the intrinsic impact of individual data points, but to establish a fair compensation scheme for data contributors. In practical settings, model developers rarely seek one single data point; instead, they acquire data in clusters or domains.
>
> **Q1**: For example, when choosing the best data mixture in the pretraining time, one can always fit a small proxy model to get the data mixture scaling law [1], where the impact of each data domain on the model performance is straightforward.
>
> **Q2**: Yes, we have provided an additional example in Appendix D.2 Data Mixture Market, where we examine acquiring data from multiple domains for pretraining. This setting, however, does not implement curriculum pricing—the pricing applies to the full pretraining run as a whole. We agree that understanding the role of curriculum data is important for data selection, but it is not clear that compensation itself must follow a curriculum structure.
>
> If you have any further questions or ideas, please let us know. We’d be glad to share additional supporting information.

---

> > ### Comment · Reviewer_avQc · 2025-11-24
> > **Thank you for the response**
> >
> > I thank the authors for their response.
> >
> > About the assumption of buyers' valuation, I agree with the author response that this can be approximated with specialised methods for different applications (e.g., LLM as judge for RAG, fitting proxy model to get scaling laws for data mixture, etc.).
> >
> > About pre-training, thanks for clarifying that the presented mechanism is more for data mixture selection rather than individual data points - it makes sense now. Thank you for updating the paper with this.
> >
> > Overall, I am satisfied with the response and will wait for discussion on the other reviews.

---

### Official Review · Reviewer_4Wcz · 2025-11-07

**Soundness:** 2
**Presentation:** 2
**Contribution:** 3
**Rating:** 4
**Confidence:** 3

**Summary:**

This paper revisits data valuation through the lens of market design, arguing that popular valuation methods such as LOO and Data Shapley are not incentive-compatible and thus lead to inefficient data markets. The authors adapt Myerson and VCG mechanisms to the data trading setting, proving that Myerson payments are the minimal truthful mechanisms optimal for buyers. Theoretical results are complemented by simulations in mean-estimation and RAG markets, showing that the proposed mechanisms maintain truthfulness and individual rationality while existing valuation methods do not.

**Strengths:**

1.	The paper provides a theoretically rigorous treatment of incentive-compatible payment mechanisms in data markets, grounded in formal mechanism design and supported by detailed proofs.
2.	The paper clearly identifies the limitations of existing valuation methods, such as LOO and Shapley, by showing their vulnerability to strategic misreporting and inefficiency in market collaboration.
3.	The paper effectively adapts classical mechanisms such as Myerson and VCG into a buyer-optimal framework and validates their practicality through an illustrative real-world RAG marketplace example.

**Weaknesses:**

1.	The paper lack analyze some related data pricing work, such as model-based data pricing[1]. This pricing approach may not present the challenges mentioned by the paper.
2.	Myerson and VCG mechanisms can be expensive, but the article has no discussion of approximate or scalable variants.
3.	The known and continuous buyer valuation assumption is unrealistic for real-world ML markets; the impossibility result for private valuations reduces generalizability.
[1] Chen L, Koutris P, Kumar A. Towards model-based pricing for machine learning in a data marketplace[C]//Proceedings of the 2019 international conference on management of data. 2019: 1535-1552.

**Questions:**

1. Could the Myerson or VCG payments be approximated efficiently for large-scale data markets?
2. In the RAG experiment, how consistent are the payment outcomes across different LLM judges or domains?
3.Would the same results hold if multiple buyers had private valuations (i.e., two-sided uncertainty)?

---

> ### Author Response · Authors · 2025-11-21
>
> Thank you for the helpful feedback. We address each point below, using Wi for weakness i and Qi for question i
>
> **W1**: Thanks for pointing this out. We have updated our manuscript to include the related works on data pricing.
>
> **W2**: Exact dominant-strategy truthfulness is extremely brittle: once payments deviate from the precise VCG/Myerson formula, the exact truthfulness is usually lost. One can, however, get ε-truthfulness / approximate IC with approximate payments. This is beyond the scope of our study. Here we just want to point out that popular data valuation methods are not suitable for data pricing due to the market dynamics.
>
> **W3**: We acknowledge that the known buyer’s valuation is rather unrealistic. However, we would like to highlight that this valuation function can be effectively approximated. For example, when choosing the best data mixture in the pretraining time (please see our Appendix D.2 Data Mixture Market), one can always fit a small proxy model to get the data mixture scaling law [1].  In the RAG case, this can be modelled with LLM as a judge, as shown in our paper. We further checked that the LLM-as-a-judge result aligns with the ground truth values, see evidence in Appendix D.3.3. We will make sure that this is clarified in our updated version.
>
> [1] Jiasheng Ye et al. Data Mixing Laws: Optimizing Data Mixtures by Predicting Language Modeling Performance
>
> **Q1**: We would like to point out that in a large-scale market with discrete W, the calculation of VCG/Myerson payment is indeed efficient. This can be seen in Line 471-476.
>
> **Q2**: The payment dynamics appear consistent across domains—our tests in legal settings produced pricing patterns similar to those observed in the medical domain. What makes a difference is the capability of the judge model. We used DeepSeek-R1 in our experiments. When RAG is needed in highly specialized areas such as medicine or law, the judge must have enough domain awareness to reliably evaluate the resulting answers. We also evaluated several newer open-weight judge models and measured the correlation between their scores and those from DeepSeek-R1. A correlation value near +1 indicates strong agreement, and our results suggest that the judge cannot be too small to maintain this consistency.
>
> | Judge model  |Qwen 2.5 - 7B | Qwen 2.5 - 72B |Qwen 3 - 235B |  Llama 3.3 - 70 B | Llama 3.2 - 3B|
> |----------|----------|----------|----------|----------|----------|
> | Pearson Correlation (%) | 72.9  | 75.6  | 85.8 | 72.5  | 51.1 |
>
>
> Q3: No, we have provided an impossibility result in Appendix E. That is, there is no truthful pricing rule with a two-sided market, and even worse, the price of Anarchy can be arbitrarily bad in a two-sided market.
>
> We hope this clarifies our findings. If you feel that our response sufficiently addresses your concerns, we kindly request that you consider raising your score. If any outstanding issues remain, please let us know, and we'll be happy to provide further clarification.

---

### Author Response · Authors · 2025-11-21
**Global response**

Dear reviewers,

Thank you for your thoughtful and insightful feedback. Via this global response, we address some of the commonly raised questions:

**Unrealistic assumption on the known buyes’ valuations.**  We acknowledge that known buyer’s valuation is rather unrealistic. However, we would like to highlight that this valuation function can be effectively approximated. For example, when choosing the best data mixture in the pretraining time, one can always fit a small proxy model to get the data mixture scaling law [1] (see our Appendix D.2 Data Mixture Market as well). In the RAG case, this can be modelled with LLM as a judge, as shown in our paper. We further checked that the LLM-as-a-judge result aligns with the ground truth values, see further evidence in Appendix D.3.3.

**Inefficiency of these truthful mechanisms.** Exact dominant-strategy truthfulness is extremely brittle: once payments deviate from the precise VCG/Myerson formula, the exact truthfulness is usually lost. One can, however, get ε-truthfulness / approximate IC with approximate payments. This is however beyond the scope of our study.
Moreover, when $W$ is discrete, as noted in Lines 471--476, the integral becomes a step function and is straightforward to evaluate. Even in the continuous setting, once the buyer's valuation is known, we can approximate the integral by sampling several $\tilde{c}_j$ values from the interval $[c_j, \infty)$; increasing the number of samples improves accuracy. This is precisely the approach we used for the mean-estimation example. We will clarify this point in the revised version.

Moreover, we have updated our manuscript a bit, with the updates marked in purple. We kindly invite you to review the updated version. Thanks in advance!

Best,

Submission 20610 authors

---

### Meta-Review · Area_Chair_4VSB · 2026-01-07

**Summary:**

he reviewers have raised the following concerns:

W1. Lack of analysis of some related work on data pricing.

W2. Lack of approximations or efficient implementations to address the computational complexity of the proposed Myerson- and VCG-based methods, limiting their applicability to large-scale data markets.

W3. Lack of discussion of the practicality of the assumptions regarding known and continuous buyer valuations.

W4. Lack of integrator robustness analysis, making the stability and reliability of the results insufficiently validated.

W5. Lack of in-depth analysis of noise robustness.

W6. Lack of an intuitive illustration of the overall architecture.

W7. The RAG application mismatches the pretraining data setup of the experiment.

W8. Lack of experiments on multi-buyer data markets.

W9. Lack of clarity regarding the technical challenges and justifications of the modeling framework.

W10. Lack of clarification on setting the experiment parameter omega to 0 or 1.

Reviewer Usk1 and Reviewers 4Wcz, C5p4, and 3FgT provided negative ratings (4) before the rebuttal. Reviewer 3FgT is likely to increase their rating from 4 to 6 due to the addressing of their concerns. Reviewers 4Wcz and C5p4 are likely to maintain their ratings because their concerns in W2 and W8 remain outstanding. Reviewer s6hu explicitly indicated maintaining their rating at 6.

Overall, this paper is borderline, and extensions to address scalability and additional experiments in multi-buyer settings could be incorporated before final acceptance. Therefore, this paper is not recommended for acceptance at this time.

**Reviewer Concerns:**

During the rebuttal, the authors provided further clarifications and discussions, as well as new experimental results and analyses, to address the raised concerns. In particular, the authors fully addressed W1, W3, W4–W7, and W9–W10. However, the rebuttal did not fully address the crucial scalability concern in W2, which was shared by Reviewers 4Wcz and C5p4, nor the multi-buyer validation concern in W8.

**Reviewer Scores:**

Reviewer Usk1 and Reviewers 4Wcz, C5p4, and 3FgT provided negative ratings (4) before the rebuttal. Reviewer 3FgT is likely to increase their rating from 4 to 6 due to the addressing of their concerns. Reviewers 4Wcz and C5p4 are likely to maintain their ratings because their concerns in W2 and W8 remain outstanding. Reviewer s6hu explicitly indicated maintaining their rating at 6.

---

### Decision · Program_Chairs · 2026-01-26

Reject